# Using a multi-layer snow model for transient paleo studies: surface mass balance evolution during the Last Interglacial

Thi-Khanh-Dieu Hoang[1], Aurélien Quiquet[1], Christophe Dumas[1], Andreas Born[2], and Didier M. Roche[1,3]

[1]Laboratoire des Sciences du Climat et de l'Environnement, LSCE/IPSL, CEA-CNRS-UVSQ, Université Paris-Saclay, 91191 Gif-sur-Yvette, France
[2]Department of Earth Sciences, University of Bergen and Bjerknes Centre for Climate Research, Bergen, Norway
[3]Department of Earth Sciences, Faculty of Science, Vrije Universiteit Amsterdam, Amsterdam, the Netherlands

**Correspondence:** Thi-Khanh-Dieu Hoang (dieu.hoang@lsce.ipsl.fr)

**Abstract.** During the Quaternary, ice sheets experienced several retreat-advance cycles, strongly influencing climate patterns. In order to properly simulate these phenomena, it is preferable to use physics-based models instead of parameterizations to estimate surface mass balance (SMB), which strongly influences the evolution of the ice sheet. To further investigate the potential of these SMB models, this work evaluates BESSI (BErgen Snow Simulator), a multi-layer snow model with high computational efficiency, as an alternative to providing SMB for the Earth system model *i*LOVECLIM for multi-millennial simulations as in paleo studies. We compared the behaviors of BESSI and ITM - Insolation Temperature Melt, an existing SMB scheme of *i*LOVECLIM during the Last Interglacial (LIG). First, we validate the two SMB models using the regional climate model MAR (Modèle Atmosphérique Régional) as forcing and reference for the present-day climate over Greenland and Antarctic Ice Sheets. The evolution of SMB over the Last Interglacial period (LIG) (130-116 kaBP) is computed by forcing BESSI and ITM with transient climate forcing obtained from *i*LOVECLIM for both ice sheets. For present-day climate conditions, both BESSI and ITM exhibit good performance compared to MAR despite a much simpler model setup. While BESSI performs well for both Antarctica and Greenland for the same set of parameters, the ITM parameters need to be adapted specifically for each ice sheet. This suggests that the physics embedded in BESSI allows better capture of SMB changes across varying climate conditions, while the ITM displays a much stronger sensitivity to its tunable parameters. The findings suggest that BESSI can provide more reliable SMB estimations for the *i*LOVECLIM framework to improve the model simulations of the ice sheet evolution and interactions with climate for multi-millennial simulations.

## 1 Introduction

The Quaternary (since 2.6 Ma) has experienced several glacial-interglacial cycles. These episodic periods influenced the whole Earth system, with climate shifting periodically from cold to warm phases and repeated retreat-advance cycles of the ice sheets and glaciers. Ice sheets and their interactions with climate strongly influence phenomena such as sea level evolution (Dutton et al., 2015; Spratt and Lisiecki, 2016; Turney et al., 2020) or changes in the atmospheric circulation (Ullman et al., 2014; Liakka et al., 2016). Ice sheets gain mass through surface accumulation (snow and rain) and internal accumulation (refreezing).

In contrast, they lose mass due to melting and sublimation/evaporation processes on the surface or through iceberg calving and sub-shelf melting. The difference between mass gains and losses at the surface is called surface mass balance (SMB), which plays a significant role in the build-up or disappearance of the ice sheets. Studies of ice sheet evolution through past events unravel the dynamics of glaciation and deglaciation, improving trajectories of ice sheets in the past as well as confidence in future projections.

Investigating ice sheets and climate feedbacks in such long-timescale periods requires a tool that can simulate the interactions between the main components of the Earth system at a reasonable computational cost. In this context, Earth system models of intermediate complexity (EMICs) are of interest as they have much lower computational costs compared to state-of-the-art general circulation models (GCMs) while still being able to simulate most of the important processes thanks to their low resolution and simplifications (Claussen et al., 2002; Eby et al., 2013). However, these simplifications result in some drawbacks, particularly in reproducing the evolution of ice sheets. Because of their coarse resolution, EMICs fail to capture the narrow ablation zones in the ice sheets' margin, leading to improper runoff estimation (Ettema et al., 2009; Noël et al., 2019). To mitigate this problem, the output of the atmospheric part can be bi-linearly interpolated (Gregory et al., 2012) or downscaled (Quiquet et al., 2021) to provide finer resolution input to the ice sheet model in the EMICs framework.

Another problem is the missing physical snow models within the EMICs framework to simulate the energy and mass transfer between the surface and the atmosphere (Lenaerts et al., 2019). In general, EMICs mostly utilize simple parameterizations such as positive degree day (PDD) (Reeh, 1991) or insolation temperature melt equation (ITM) (Van Den Berg et al., 2008) due to their simplicity and low computational cost (Born and Nisancioglu, 2012; Stone et al., 2013; Robinson and Goelzer, 2014; Goelzer et al., 2016b; Quiquet et al., 2021). However, as these schemes depend on locally calibrated parameters, their reliability is questioned when climate conditions change or when available data for calibration is limited, particularly in paleo studies. Bauer and Ganopolski (2017) report a failure of PDD in providing proper SMB values for the last glacial cycle study, which resulted from the albedo feedback being absent in the simulation. This poses a need to include a more physical snow model in such long-term climate simulations. The first option is to use dedicated snowpack models coupled to regional climate models (RCMs), which have abilities to simulate not only the key physical processes of SMB (melt, sublimation, and snow drifting) but also snow properties such as densities and metamorphism (Fettweis et al., 2017; Noël et al., 2018; Agosta et al., 2019; van Dalum et al., 2022). However, due to their complexity and computational cost, they are not suitable for long-term transient simulations and large study areas. As a compromise between parameterizations and SEB models, intermediate complexity energy balance models are promising SMB schemes for EMICs to run long simulations of ice sheet studies (Calov et al., 2005; Willeit et al., 2024). These models have the appropriate level of simplicity in their structure and high computational efficiency, such as Born et al. (2019).

To answer the question of whether a physics-based scheme is a better choice for the representation of SMB for paleo timescale, this work evaluates the differences in the behaviors of the simple SMB scheme in *i*LOVECLIM and a physical-based surface energy balance model BESSI (Bergen Snow SImulator) (Born et al., 2019) in a paleo study. Thanks to its high computational efficiency, *i*LOVECLIM has been used to carry out many paleoclimate studies ranging from ice sheet-climate interactions during the last deglaciation (Roche et al., 2014a; Quiquet et al., 2021; Bouttes et al., 2023), Heinrich

Events (Roche et al., 2014b), to ocean circulation (Lhardy et al., 2021a) and carbon cycle changes between glacial-interglacial states (Bouttes et al., 2018; Lhardy et al., 2021b). BESSI is a surface energy and mass balance model designed for Earth system models of intermediate complexity. The snow model has been used to study the surface mass balance of the Greenland ice sheet during different periods (Zolles and Born, 2021; Holube et al., 2022; Zolles and Born, 2022) and proved to have good performance compared to other more complex models (Fettweis et al., 2020). In this work, we evaluate the performance of the updated version of BESSI since Zolles and Born (2021) and ITM - the current SMB scheme of *i*LOVECLIM for present-day climate using the regional climate model MAR (Modèle Atmosphérique Régional) as forcing and benchmark in Greenland and Antarctic Ice Sheets (GrIS and AIS, respectively). By doing this, we assess the models' behaviors under different climate conditions. In the second part, we assess the impact of using *i*LOVECLIM as the climate forcing on the SMB simulation of BESSI and ITM. Next, we compare the SMB evolution simulated by the two SMB models during the most recent interglacial period (LIG) (130-116 kaBP), which corresponds to the marine isotope stage (MIS) 5e. During this period, due to the change in the orbital configuration of the Earth, increasing summer insolation in the high latitude of the Northern Hemisphere leads to warmer conditions in polar regions (Capron et al., 2014). The estimation of the global mean temperature change during the LIG with respect to the pre-industrial ranges from almost no change (Capron et al., 2014; Hoffman et al., 2017; Otto-Bliesner et al., 2021) to a 1 to 2°C warming (Turney and Jones, 2010; McKay et al., 2011; Fischer et al., 2018). A warming in the high-latitude regions is nonetheless reported by both proxy data and model outputs. In addition, the sea level is reported to be at least 1.2 meters higher during the LIG (Dutton and Lambeck, 2012; Dutton et al., 2015; Dyer et al., 2021). Hence, the LIG provides documented records and insights into the behaviors of different Earth system components under warm climates to benchmark models and study the dynamics behind the phenomena (Fischer et al., 2018). This period has been well-studied for various aims such as reconstructing temperature (Lunt et al., 2013; Landais et al., 2016; Obreht et al., 2022) and sea level (Kopp et al., 2013; Dutton et al., 2015); investigating climate and ice-sheet interactions (Bradley et al., 2013; Goelzer et al., 2016a; Sutter et al., 2016). Applying BESSI for the LIG has been done before in the work of Plach et al. (2018) for the Greenland Ice Sheet only by using climate forcings from MAR with equilibrium runs of some LIG time slices: 130, 125, 120, and 115 kaBP. In our work, as *i*LOVECLIM is much more computationally inexpensive compared to MAR, we can obtain transient climate forcings for BESSI and ITM to simulate the evolution of SMB throughout the whole LIG period for both GrIS and AIS. We select the LIG to investigate the abilities of BESSI and ITM in simulating the evolution of SMB under different boundary conditions (deglaciation and glacial inception). From this, we can thoroughly investigate the effects of using a more physics-based model in simulating SMB for an intermediate complexity Earth model.

Section 2 provides background information about the models, the climate forcings, together with the design of the experiments. The results are presented in section 3, followed by a discussion about the models' behaviors and the climate forcings in section 4. Finally, a summary of the work is in section 5.

## 2 Methods

### 2.1 Models description

#### 2.1.1 BESSI

BErgen Snow SImulator - BESSI is a multi-layer snow model simulating the surface energy and mass balance with high computational efficiency, designed to be coupled with low-resolution Earth system models (Born et al., 2019). The model, which in its current configuration uses 15 vertical snow layers, requires near-surface air temperature, total precipitation, humidity, surface pressure, and downward long-/short-wave radiation as input. BESSI runs at a daily time step and simulates albedo, which decays exponentially after the latest snowfall event. Based on the energy transfer between the surface and the air, the model simulates important processes of surface mass balance, such as melt, refreezing, runoff, and sublimation/evaporation, which results in the changing mass of the snow column. Among the snow layers, heat diffusion and mass compaction are also simulated (Fig. 1). Compared to the version in Zolles and Born (2021), in this work, BESSI acquires the incoming longwave radiation flux from the input instead of using parameterization. A detailed description of surface energy and mass balance processes is presented in Appendix A.

**Figure 1.** Sketch of BESSI model with required inputs and simulated processes

#### 2.1.2 *i*LOVECLIM

The Earth system model of intermediate complexity *i*LOVECLIM (version 1.1) is a code fork of the LOVECLIM 1.2 model originated from Goosse et al. (2010). The key components of the model include the modules ECBILT for the atmosphere, CLIO for the ocean, and VECODE for the vegetation. ECBILT is a quasi-geostrophic atmospheric model that runs on a T21 spectral grid (Opsteegh et al., 1998). Meanwhile, CLIO is a 3D free-surface ocean general circulation model coupled to a thermodynamic sea-ice model and discretized on $3° \times 3°$ spherical grid (Goosse and Fichefet, 1999). VECODE is a dynamical vegetation model that allocates carbon and simulates land cover and tree fraction on the same grid as the atmospheric model (Brovkin et al., 1997). *i*LOVECLIM runs with a 360-day calendar.

Climate forcings for BESSI are obtained from the online downscaling module within *i*LOVECLIM framework, which recomputes the surface energy budget and total precipitation on a subgrid resolution for the ice sheet areas (Quiquet et al., 2018).

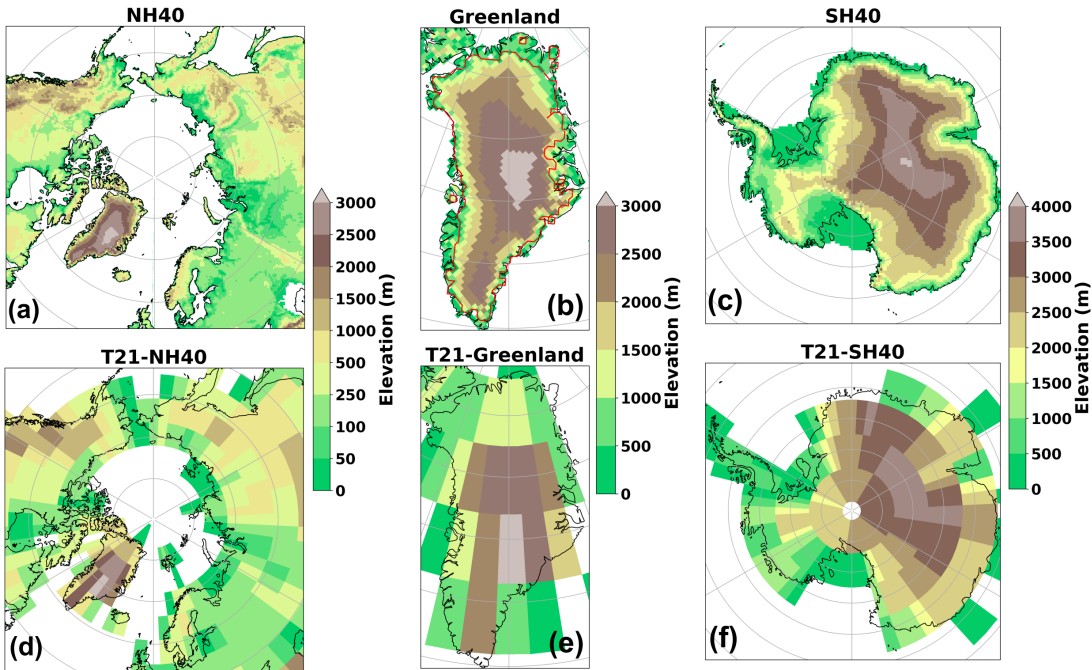

**Figure 2.** Topography of *i*LOVECLIM for different resolutions: (**a**) NH40, (**b**) NH40 zoomed in Greenland with the red contour indicates present-day ice sheet extent, (**c**) SH40, (**d**) T21 with similar projection as NH40, (**e**) Similar to (**d**) but zoomed in Greenland and (**f**) T21 with similar projection as SH40.

In this work, we run the downscaling for two polar regions to obtain near-surface air temperature, total precipitation, and humidity on a 40km×40km Cartesian grid (referred to as NH40 and SH40 for the North and South Poles, respectively) (Fig. 2a-c). To obtain other input variables for BESSI, long-/short-wave radiation, and surface pressure are bi-linearly interpolated

from the native T21 grid (Fig. 2d-f) to the NH40/SH40 grid.

In fact, due to its coarse resolution and simplification in physics, *i*LOVECLIM displays some incorrect climate patterns. Particularly, Heinemann et al. (2014) reported surface air temperature biases of *i*LOVECLIM compared to observation in North America and Northern Europe, which are preserved in the downscaling version NH40 (Quiquet et al., 2018). To evaluate the impacts of these biases on the SMB simulation, we carry out a simple bias correction process by using ERA5 (Muñoz-

120 Sabater et al., 2021), a reanalysis climate data as reference (see Appendix C). In general, the variables with strong biases are total precipitation, shortwave radiation and air temperature (Fig. C1 and C2). In addition, for the Antarctic Ice Sheet, the humidity is strongly underestimated in *i*LOVECLIM. These biases might partly come from simplified physics and the lack of explicit vertical representation in iLOVECLIM. For example, the clouds are prescribed based on the present-day climatology. These biases need further investigation in future works.

### 2.1.3 ITM stand-alone

In terms of the SMB scheme, *i*LOVECLIM uses the insolation temperature melt method (ITM) (Van Den Berg et al., 2008). This parameterization is implemented to provide SMB to the ice sheet model embedded in *i*LOVECLIM named GRISLI for coupling purposes (Quiquet et al., 2021).

This parameterization calculates the runoff water (in mWE d$^{-1}$) as

$$\frac{\partial m_{runoff}}{\partial t} = \frac{1}{\rho_w L_m}((1-\alpha_s)SW + crad + \lambda(T_{air} - 273.15)) \geq 0 \tag{1}$$

in which, $\rho_w$ is liquid water density (1000 kg m$^{-3}$), $L_m$ is the specific latent heat of melting (3.34×10$^5$ J kg$^{-1}$), $\alpha_s$ is the surface albedo, $SW$ is the surface shortwave radiation (W m$^{-2}$) and $T_{air}$ is the near-surface air temperature (K). Meanwhile, $\lambda$ and $crad$ are two empirical parameters.

For the coupling between *i*LOVECLIM and GRISLI, Quiquet et al. (2021) implemented an albedo interpolation to take into account the altitude of the grid points (vertical) and to create a smooth transition of albedo value from ocean to land area (horizontal). In addition, to take into account the temperature bias of *i*LOVECLIM, a local modification of the parameter $crad$ based on the annual mean temperature difference compared to ERA-Interim (Dee et al., 2011) is also included in ITM as explained in Quiquet and Roche (2024).

Here, to provide a clean comparison to BESSI, a stand-alone version of ITM is used with the same albedo value as the ice grid points in *i*LOVECLIM ($\alpha_s$ = 0.85) and $\lambda$ = 10 W m$^{-2}$ K$^{-1}$ as in Quiquet et al. (2021). The input data $SW$ and $T_s$ are read from BESSI input, hence, ITM also runs at a daily time step. The empirical parameter $crad$ is tuned for the present-day climate with MAR as forcing. The SMB is the remaining total precipitation (accumulation) after subtracting the calculated runoff only (ablation), with the sublimation process being neglected.

## 2.2 Present-day climate reference data

For calibration/validation purposes, we use the present-day climate data from one of the state-of-the-art regional climate models - MAR (Modèle Atmosphérique Régional). MAR has been widely applied to study the SMB changes and surface melt for polar regions (Fettweis et al., 2017; Agosta et al., 2019; Mankoff et al., 2021). The model, with a typical sub-daily time step of 120 s Fettweis (2007)), includes a 3D atmospheric model coupled with a 1D surface-atmosphere energy mass exchange scheme named SISVAT (Soil Ice Snow Vegetation Atmosphere Transfer) (Fettweis et al., 2017) that is more complex and physical than BESSI. It can simulate up to 30 layers of snow/ice and consider snow properties and metamorphism (Kittel et al., 2021). Also, the simulated surface albedo takes into account more variables, including snow's optical properties, clouds, snow depth, the presence of bare ice, and liquid water (Tedesco et al., 2016). Detailed about the MAR model and its setup can be found in Fettweis (2007) and Fettweis et al. (2013).

In this study, MAR acts as present-day forcing and reference benchmarks to compare with BESSI and ITM for both Greenland and Antarctic Ice Sheets (denoted as GrIS and AIS, respectively). The resolution of the climate forcings is 15km×15km for GrIS (version 3.13) (Fig. 3a) and 35km×35km grid for AIS (version 3.12) (Fig. 3b), covering the period 1979 - 2021.

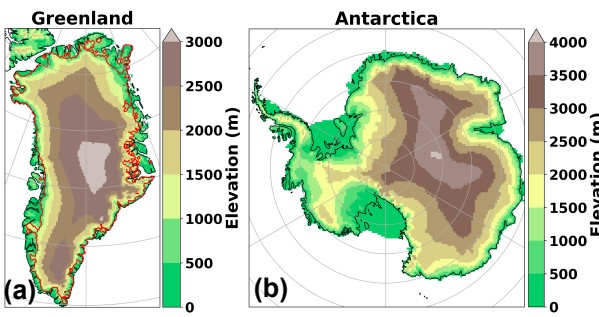

**Figure 3.** Topography of MAR for (**a**) Greenland (15km×15km) with the present-day ice sheet extent in red contour and (**b**) Antarctica (35km×35km).

## 2.3 Study design

In this work, we carry out three sets of experiments corresponding to the two climate forcings: MAR for present-day conditions and *i*LOVECLIM for both present-day and the LIG conditions. The climate characteristics of these experiments are presented

in Table 1.

In the first experiment, we investigate the behaviors of BESSI and ITM for present-day climate by using MAR as forcing (BESSI-MAR and ITM-MAR). The calibration and validation are carried out for GrIS and AIS during the study period from 1979 to 2021 with the calibration carried out for GrIS only. To evaluate the results, we use two goodness-of-fit metrics, which are the coefficient of determination $R^2$ and the Root Mean Squared Errors (RMSE) to assess the differences of BESSI-MAR

and ITM-MAR refer to MAR (see Appendix B). Initially, BESSI is spun up by looping the forcing several times until it reaches an equilibrium state. The ice mask corresponds to present-day ice sheet extent, classified in MAR as grid cells with more than 50% of permanent ice. Some of BESSI's parameters related to albedo simulation ($\alpha_{freshsnow}$, $\alpha_{firn}$, and $\alpha_{ice}$) and turbulent latent heat flux calculation ($r_{lh/sh}$ and $D_{sh}$) are tuned to obtain lowest RMSE value between BESSI and MAR output and the narrowest gap in term of total SMB (SMB integrated over the ice sheet mask). The final values of these parameters are

presented in Table A1. The same tuning procedure is applied for the empirical parameter $crad$ of ITM, and the optimized value is -10 W m$^{-2}$.

Before applying BESSI and ITM for the LIG with *i*LOVECLIM as forcing, we investigate the influences of the input on the behavior of the two SMB models by comparing the results of BESSI-*i*LOVECLIM and ITM-*i*LOVECLIM to MAR for the present-day condition. *i*LOVECLIM forcings for present-day are obtained by running the model with the prescribed

greenhouse gases (GHG) concentrations during the same period as MAR, from 1979 to 2021. As mentioned above (Sec 2.1.2), we implement a simple bias correction process to correct the climate field of *i*LOVECLIM. To quantify the impact of these biases on the SMB simulation, in addition to original climate forcings, we also run BESSI and ITM with the bias-corrected version of *i*LOVECLIM.

**Table 1.** Climate characteristics of two different climate forcings: MAR and *i*LOVECLIM for different experiments. The calibration/validation is carried out from 1979 to 2021 with forcings from MAR and iLOVECLIM. Mean summer shortwave radiation and mean summer temperature are calculated based on the present-day ice sheet extent. The climate forcings for the Last Interglacial (LIG) comes from *i*LOVECLIM only. The summer insolation of the paleo study corresponds to the summer insolation of 65°N for the Greenland Ice Sheet (GrIS) and 65°S for the Antarctic Ice Sheet (AIS). The summer months are June-July-August for GrIS and December-January-February for AIS.

| Present-day climate (1979 - 2021) | | | | |
|---|---|---|---|---|
| Climate forcings | MAR | | *i*LOVECLIM | |
| Ice sheet | GrIS | AIS | GrIS | AIS |
| Mean summer shortwave radiation (W m$^{-2}$) | 289.95 | 322.58 | 48.01 | 48.48 |
| Mean summer temperature (°C) | -7.67 | -19.9 | -4.14 | -17.77 |

| Paleo study | | | | |
|---|---|---|---|---|
| Period | PI | | LIG | |
| Ice sheet | GrIS | AIS | GrIS | AIS |
| Carbon dioxide (ppm) | 280.00 | | 202.61 to 283.03 | |
| Summer insolation (W m$^{-2}$) | 475.19 | 507.12 | 437.68 to 540.93 | 460.23 to 517.6 |
| Mean summer temperature (°C) | -4.63 | -18.52 | -8.22 to -0.41 | -20.79 to -18.09 |
| Global mean temperature (°C) | 15.89 | | 14.89 to 16.6 | |

For the LIG, to obtain the climate forcing, we run *i*LOVECLIM transiently from 135 to 115 kaBP, with present-day ice sheet topography and varying orbital configuration and concentrations of GHG. For every 500 years, we sample 50 years of daily output to provide forcings for BESSI and ITM. In total, there are 41 sets of inputs corresponding to 41 timeslices covering the entire LIG period. BESSI is spun up with the input data from the first time slice - 135 kaBP to reach the equilibrium state. Then, for each time slice, we run BESSI for 100 years with the snowpack from the spin-up and take the annual mean of the last 50 years for further analysis. The evolution of SMB simulated by BESSI and ITM during the LIG is then compared to investigate the models' behaviors. In order to assess the trend of SMB evolution, we compute the differences in the annual mean SMB during the LIG with respect to the pre-industrial (PI) value for both BESSI-*i*LOVECLIM and ITM-*i*LOVECLIM. The climate forcing of PI is obtained by running downscaled *i*LOVECLIM for 50 years from a 1000-year spin-up under pre-industrial boundary conditions. To quantify the biases of climate forcings on the models' behaviors, assuming the biases in *i*LOVECLIM are constant with time, we use the present-day bias correction factors to correct the climate forcings for LIG and PI. The results of before and after the bias correction are then compared.

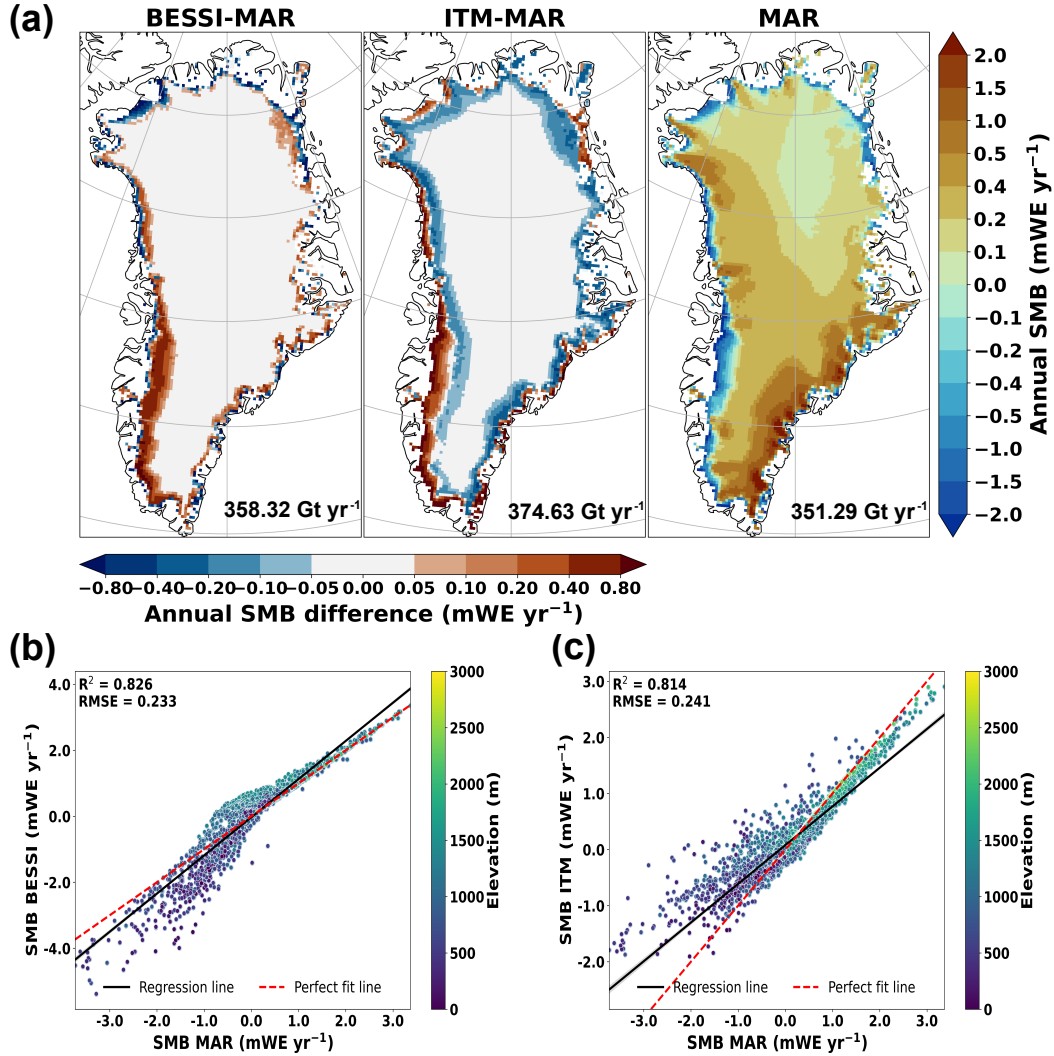

**Figure 4.** (**a**) Annual mean SMB anomalies (in mWE yr⁻¹) of BESSI-MAR and ITM-MAR compared to MAR for Greenland Ice Sheet. The reference, MAR, is shown in absolute annual SMB values. The total SMB (in Gt yr⁻¹) integrated for the ice sheet area is also included. The scatter plots of (**b**) BESSI-MAR vs. MAR and (**c**) ITM-MAR vs. MAR indicate the SMB of each grid point (in mWE yr⁻¹) with elevation classification, including the linear regression line in black and the perfect fit line (1,1) in red.

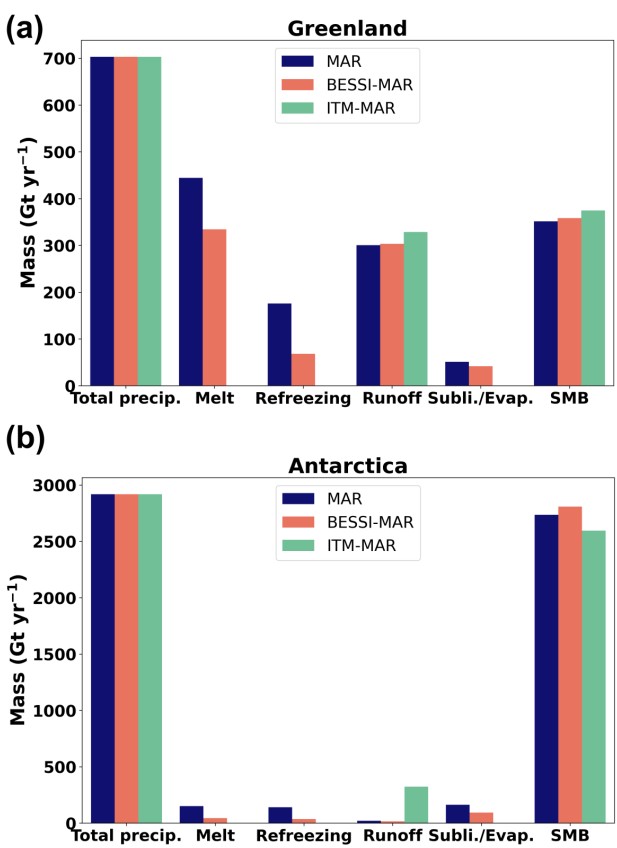

**Figure 5.** Contribution of different key processes to the 43-year mean total SMB of MAR, BESSI-MAR and ITM-MAR in (**a**) Greenland and (**b**) Antarctica (in Gt yr$^{-1}$).

## 3  Results

### 3.1  MAR as present-day climate forcings

#### 3.1.1  Greenland

The map of the annual mean SMB differences simulated by BESSI-MAR and ITM-MAR compared to MAR (shown in absolute
value) for the Greenland ice sheet during the period 1979 - 2021 is presented in Fig. 4a. For BESSI-MAR, in the southwest of
Greenland, there is a widespread of positive SMB anomalies, indicating an underestimation of this ablation zone which is also
reported by Plach et al. (2018); Fettweis et al. (2020) (Supplementary Fig. S1). Such high SMB values in BESSI-MAR for this
area is related to the albedo simulation. Compared to MAR, the annual albedo simulated for the southwest of GrIS is higher in
BESSI-MAR, leading to a lower runoff rate (Supplementary Fig. S2). Even though the extent is underestimated, the magnitude
of ablation in BESSI-MAR is higher than MAR around the margins, particularly in the North and West of Greenland. For

these grid points, BESSI-MAR simulates high melt rates while the amount of water refreeze remains low (Supplementary Fig. S3a-b), resulting in negative SMB anomalies compared to MAR. In the center of the ice sheet, where sublimation/evaporation is dominant due to dry climate, the SMB is simulated correctly by BESSI-MAR as referred to MAR. However, this process is slightly underestimated in some areas, noticeably the west of the ice sheet (Supplementary Fig. S3c). In general, the 43-year mean SMB simulated by BESSI-MAR is in good agreement with MAR despite a simpler model structure with a 2% difference in the total SMB.

For ITM-MAR, the differences in SMB compared to MAR come from the runoff simulation, as the model does not simulate other processes. Hence, the differences are located mostly in low elevation areas where the temperature is not low enough to compensate for the shortwave radiation influence (Eq. (1)) during the summer months (Supplementary Fig. S4a-b). Around the ice sheet margin, ITM-MAR simulates less runoff around the margins due to a constant albedo value (0.85) (Supplementary Fig. S2), resulting in SMB overestimation for these grid points. The total SMB difference between ITM-MAR and MAR is around 6.64%, three times more than that of BESSI-MAR.

The scatter plots of the grid points with different elevations in the SMB maps are also presented in Fig. 4, with the evaluation metrics to illustrate the goodness-of-fit of BESSI-MAR and ITM-MAR to MAR. Compared to MAR, BESSI-MAR tends to underestimate SMB of the low-elevation grid points located in the ice sheet margin in the North and West (Fig. 4b). For points located near the equilibrium line (with SMB $\approx$ 0), SMB is slightly overestimated in BESSI-MAR. Meanwhile, ITM-MAR shows a trend of SMB overestimation for grid points located in the ablation area (Fig. 4c). In general, the evaluation metrics illustrate an acceptable SMB simulation of both BESSI-MAR and ITM-MAR with respect to MAR.

Fig.5a illustrates the mean value of total SMB elements simulated by the three models for GrIS. For BESSI-MAR, we can see strong underestimations in melt and refreezing compared to MAR, especially refreezing with less than half of MAR's value. This might result from the daily time step, which causes the model to neglect the diurnal temperature cycle (Krebs-Kanzow et al., 2018). However, these underestimations are compensated in the runoff, leading to an acceptable value in BESSI-MAR compared to MAR. Meanwhile, the sublimation/evaporation rate in BESSI-MAR is slightly lower than MAR due to the underestimation of this process. For ITM-MAR, the model compensates for the absence of the sublimation/evaporation process by simulating more runoff to obtain a similar SMB rate compared to MAR. Both BESSI-MAR and ITM-MAR overestimate the SMB with MAR as a reference. This trend is consistent during the study period (Supplementary Fig. S5a).

### 3.1.2 Antarctica

The annual mean SMB differences of BESSI-MAR and ITM-MAR with respect to MAR for the Antarctic Ice Sheet are shown in Fig. 6a. For Antarctica, BESSI-MAR shows a high agreement with MAR on the SMB simulation with very limited differences. The problem related to melting in Greenland is limited here as it has a much colder climate (Supplementary Fig. S6a), and sublimation/evaporation becomes dominant. The differences between the two models come from the underestimation of sublimation/evaporation around the ice sheet margin in BESSI-MAR (Supplementary Fig. S6b), leading to the larger gap between BESSI-MAR and MAR for this process compared to GrIS (Fig. 5)

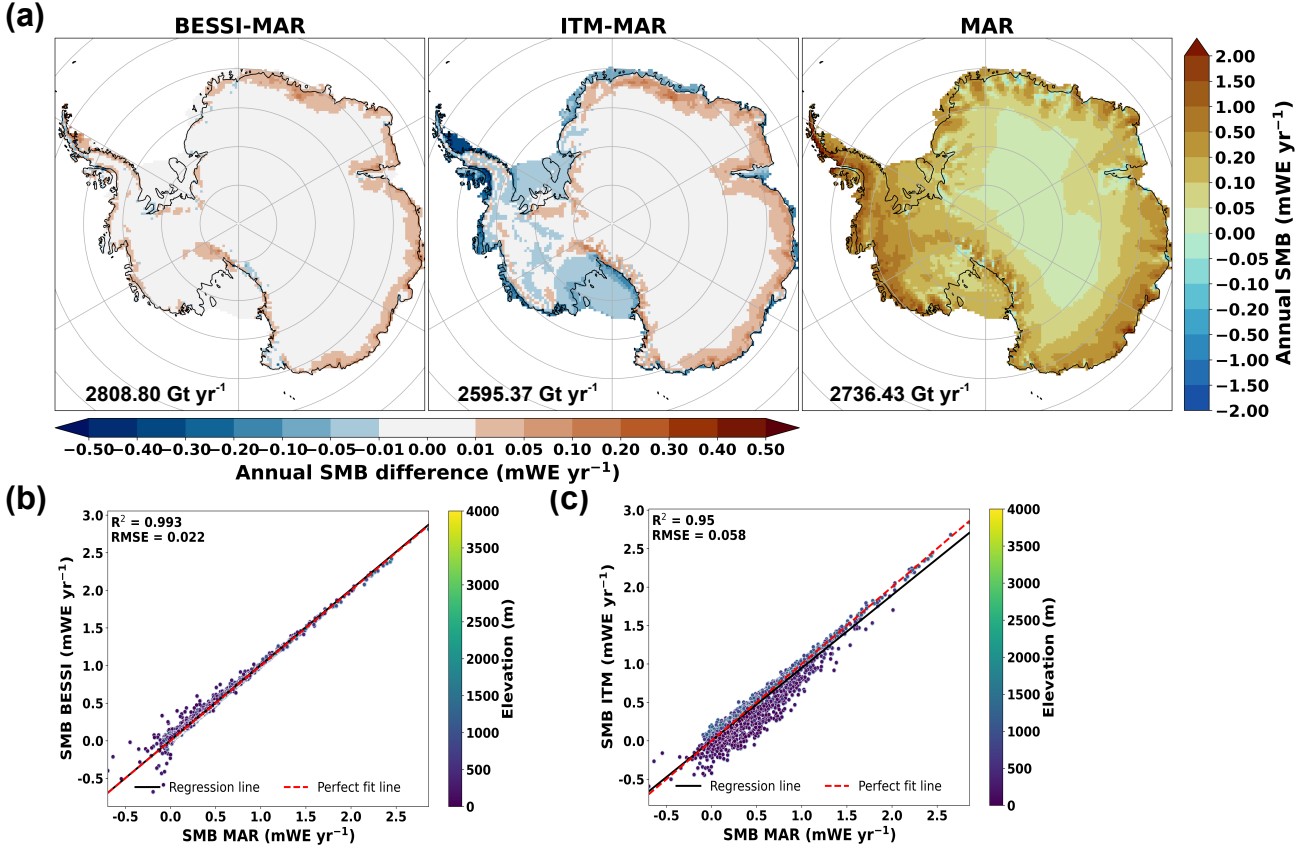

**Figure 6.** (**a**) Annual mean SMB anomalies (in mWE yr⁻¹) of BESSI-MAR and ITM-MAR compared to MAR for Antarctic Ice Sheet. The reference, MAR, is shown in absolute annual SMB values. The total SMB (in Gt yr⁻¹) integrated for the ice sheet area is also included. The scatter plots of (**b**) BESSI-MAR vs. MAR and (**c**) ITM-MAR vs. MAR indicate the SMB of each grid point (in mWE yr⁻¹) with elevation classification, including the linear regression line in black and the perfect fit line (1,1) in red.

Meanwhile, ITM-MAR exhibits large differences from MAR for the annual mean SMB. The anomalies located in the interior
of the ice sheet come from the absence of sublimation/evaporation in this parameterization. The underestimation of SMB around the edge of the ice sheet and the ice shelves comes from the high simulated runoff by ITM-MAR (Supplementary Fig. S6a). ITM-MAR simulates runoff for these grid points due to high shortwave radiation that overweights the mild temperature during the melting season (Supplementary Fig. S7a-b). In terms of total SMB, the differences between the two SMB models compared to MAR are in an acceptable range: 2.64% for BESSI-MAR and -5.15% for ITM-MAR (Fig.5b).

Similar to GrIS, scatter plots of the grid points with different elevations in the maps of Fig. 6a are also presented in Fig. 6b-c. For AIS, there is no significant trend of under-/over-estimation of annual mean SMB in BESSI-MAR compared to MAR (Fig. 6b). On the other hand, ITM-MAR shows a strong SMB underestimation trend for low elevation grid points (Fig. 6c) due to high runoff rates. These points correspond to the ablation zone over ice shelves that is not present in MAR. This trend is

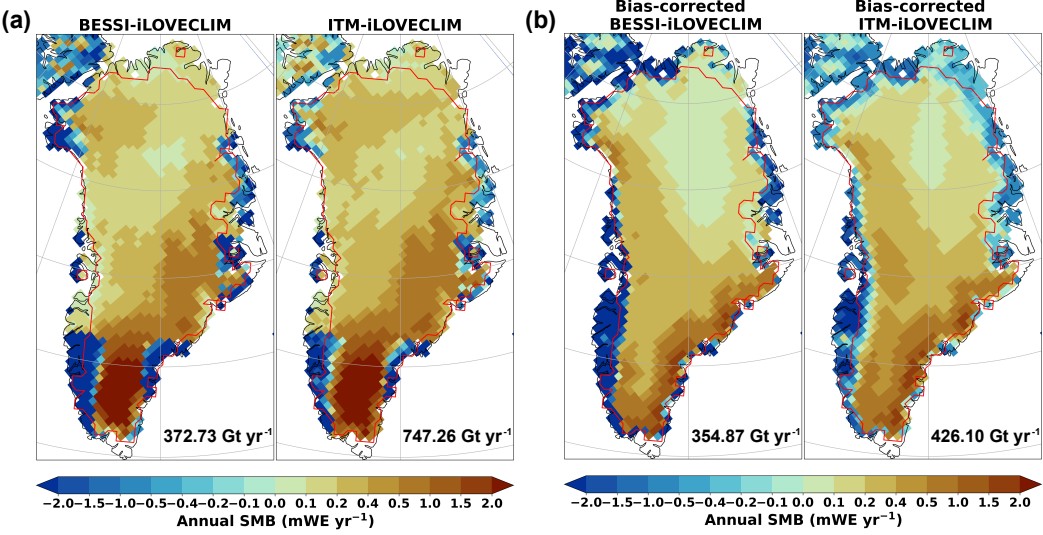

**Figure 7.** Comparison of annual mean SMB (in mWE yr[-1]) between BESSI-*i*LOVECLIM and ITM-*i*LOVECLIM (**a**) before and (**b**) after bias correction for Greenland Ice Sheet. The total SMB (in Gt yr[-1]) integrated for the present-day ice sheet extent (red line) is also included.

observed throughout the study period (Supplement Fig. S5b). The evaluation metrics suggest a good fit of the two SMB models
to MAR, with BESSI-MAR having a slightly better value.

## 3.2 *i*LOVECLIM as climate forcing: present-day

### 3.2.1 Greenland

*i*LOVECLIM has a coarser resolution and simpler model setup than MAR - a state-of-the-art regional climate model used to calibrate/validate BESSI and ITM. This difference in the simulated climate strongly influences the behaviors of the two
SMB models. Annual mean SMB during the period 1979-2021 simulated by BESSI-*i*LOVECLIM and ITM-*i*LOVECLIM for GrIS is presented in Fig. 7a. Switching the climate forcings, the resolution of *i*LOVECLIM influences BESSI-*i*LOVECLIM significantly with the SMB patterns following the input fields grid (Supplement Fig. S4). Particularly, compared to BESSI-MAR for the same study period (Supplement Fig. S1), the narrow ablation zones in the southwest of GrIS is missing while there are larger ablation zones in the South. Also, the magnitude of negative SMB in BESSI-*i*LOVECLIM is very high. This
results from a warm climate that induces high melt rates, while the model does not simulate the refreezing process well due to a large time step (as mentioned in Sect. 3.1.1). Consequently, the contribution of runoff to the total SMB in BESSI-*i*LOVECLIM are very high compared to MAR as illustrated in Fig. 8a, leading to similar SMB value even for higher total precipitation rate (372.73 Gt yr[-1] vs 351.29 Gt yr[-1], respectively). On the other hand, due to the drier atmosphere (Supplement Fig. S4d), the sublimation/evaporation in BESSI-*i*LOVECLIM is around 30% higher than in MAR.

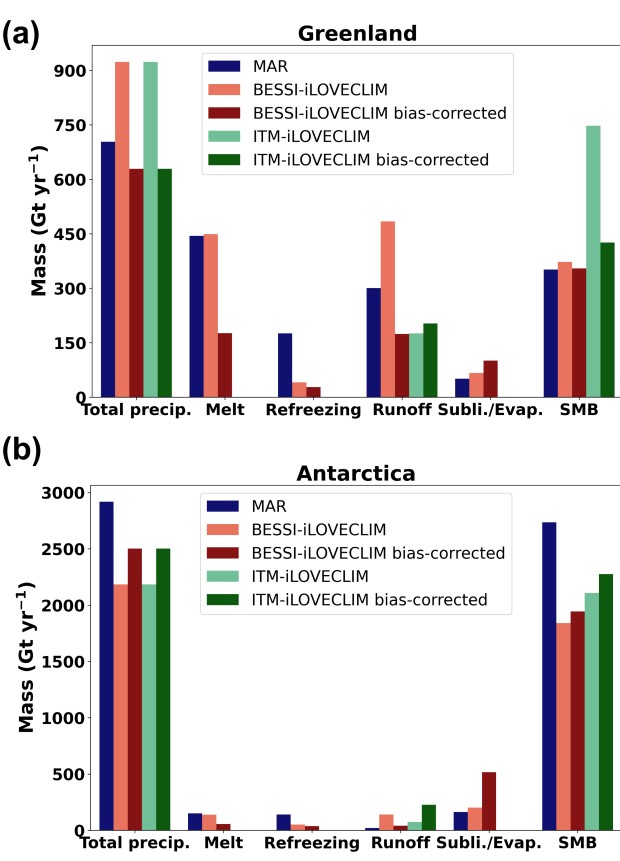

**Figure 8.** Comparison of the contribution of different key processes to the 43-year mean total SMB of BESSI-*i*LOVECLIM and ITM-*i*LOVECLIM before and after bias correction in (**a**) Greenland and (**b**) Antarctica with MAR as reference (in Gt yr$^{-1}$).

The climate forcing also strongly influences ITM-*i*LOVECLIM with similar large ablation zones in the South of GrIS as in BESSI-*i*LOVECLIM. However, the magnitude of negative SMB in ITM-*i*LOVECLIM is not as large as in BESSI, which is a result of the low shortwave radiation rates in this climate forcing (Table 1 and Supplement Fig. S4b). Therefore, the runoff contribution to the total SMB for ITM-*i*LOVECLIM is lower than MAR (Fig. 8a). For a higher total precipitation rate, this results in a much higher SMB value as indicated in Fig. 7a.

As the biases in *i*LOVECLIM exhibits a strong influence on BESSI and ITM, the annual mean SMB simulated with a corrected climate forcing is presented in Fig. 7b. With the adjusted input, BESSI-*i*LOVECLIM simulates more appropriate SMB patterns with the narrow ablation zone in the southwest presence and a bigger extent of the low accumulation zone in the center North of the ice sheet. For ITM-*i*LOVECLIM, similar patterns are observed with additional ablation zones in the North as in ITM-MAR (Supplement Fig. 1), resulting from high shortwave radiation rates in these grid points (Fig. C1). The

contribution of difference processes to the total SMB of bias-corrected BESSI-*i*LOVECLIM and ITM-*i*LOVECLIM are shown

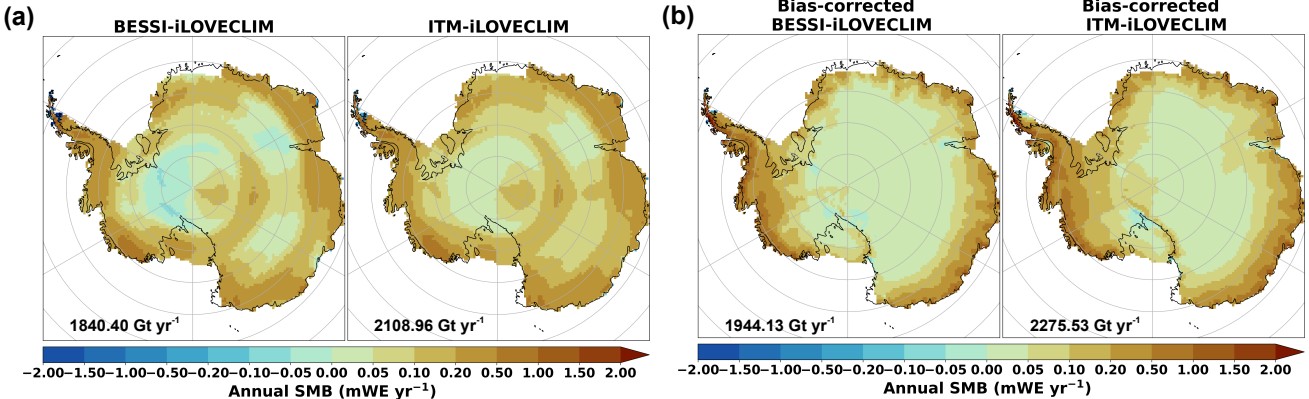

**Figure 9.** Comparison of annual mean SMB (in mWE yr⁻¹) between BESSI-*i*LOVECLIM and ITM-*i*LOVECLIM (**a**) before and (**b**) after bias correction for Antarctic Ice Sheet. The total SMB (in Gt yr⁻¹) integrated for the present-day ice sheet extent is also included.

in Fig. 8a together with results from MAR and original *i*LOVECLIM. Noticeably, the total precipitation after the bias correction in *i*LOVECLIM decreases from 923 Gt yr⁻¹ to 629 Gt yr⁻¹, around 10.5% lower than MAR's value (703 Gt yr⁻¹). This is the result of limiting the correction factors to be in the range of 0.1 to 10.0, neglecting extreme values. For BESSI-*i*LOVECLIM, as the climate is cooler after the bias correction, the runoff rate reduces to 174 Gt yr⁻¹, nearly three times less than before the

bias correction (487 Gt yr⁻¹). Because of the colder climate, the sublimation/evaporation rate increases from 67 Gt yr⁻¹ to 101 Gt yr⁻¹, nearly double the MAR's value (51 Gt yr⁻¹). For ITM-*i*LOVECLIM, the simulated runoff increases slightly from 176 Gt yr⁻¹ to 203 Gt yr⁻¹ after the bias correction, which is due to the higher shortwave radiation rates. Since there is a reduction in the total precipitation, the total SMB in ITM-*i*LOVECLIM also declines to 426.10 Gt yr⁻¹ from 747.26 Gt yr⁻¹ (around 43 %), as shown in Fig. 7. The results indicate the importance of the climate forcings quality on the results of the two SMB models.

**3.2.2 Antarctica**

For AIS, the annual mean SMB from 1979-2021 simulated by BESSI-*i*LOVECLIM and ITM-*i*LOVECLIM is presented in Fig. 9a. Similar to GrIS, the patterns of climate fields, mostly total precipitation (Supplement Fig. S7), strongly influence the simulated SMB by the two SMB models. Noticeably, there are large ablation zones observed in the center West and some parts of the East of the ice sheet in BESSI-*i*LOVECLIM, caused by the very low humidity (Fig. C2 and Supplement Fig. S7d). As

shown in Fig. 8b, this bias leads to unrealistic sublimation/evaporation simulation by BESSI-*i*LOVECLIM, around 25% higher than in MAR (around 202 Gt yr⁻¹ compared to 162 Gt yr⁻¹). Fig. 8b also indicates a low total precipitation rate of only 2184 Gt yr⁻¹ in *i*LOVECLIM, nearly 25% lower than in MAR (2919 Gt yr⁻¹). The total SMB simulated by BESSI-*i*LOVECLIM for this ice sheet is around 1840.4 Gt yr⁻¹, around 33% lower than in MAR (2736.43 Gt yr⁻¹). Meanwhile, the total SMB simulated by ITM-*i*LOVECLIM is 2108.96 Gt yr⁻¹. This rate is slightly higher than in BESSI-*i*LOVECLIM and around 23%

lower than MAR's value. Because of the low values of shortwave radiation and summer temperature, the contribution of runoff

to the total SMB in ITM-*i*LOVECLIM for this ice sheet is relatively low, which is only 75 Gt yr$^{-1}$ compared to 141 Gt yr$^{-1}$ of BESSI-*i*LOVECLIM.

The annual mean SMB simulated by BESSI and ITM with bias-corrected *i*LOVECLIM for AIS is shown in Fig. 9b. For both the two models, the SMB patterns improve significantly with the corrected climate forcings. In BESSi-*i*LOVECLIM, the widespread ablation zones are removed. However, the bar chart indicates that the sublimation/evaporation in BESSI-*i*LOVECLIM is nearly two times higher after the bias correction (Fig. 8b). This is because of the colder climate as the temperature decreases while the humidity around the margin remains low after the bias correction (Fig. C2). For ITM-*i*LOVECLIM, the larger values of shortwave radiation around the ice sheet edge induce a three times higher runoff rate (Fig. 8b). Such a high runoff contribution is also observed before in ITM-MAR (Fig. 5). Despite the bias correction, the total precipitation in *i*LOVECLIM remains below MAR's value due to the restriction range of the bias correction factor (see Appendix C). The gap is about 417 Gt yr$^{-1}$, which is around 14% of the total precipitation in MAR. This leads to lower total SMB rates in both BESSI-*i*LOVECLIM and ITM-*i*LOVECLIM in comparison with MAR, with the difference is nearly -29% in BESSI and around -17% in ITM.

### 3.3  *i*LOVECLIM as climate forcing: Last Interglacial

#### 3.3.1  Climate of the Last Interglacial

The external forcings of *i*LOVECLIM, including the summer insolation of 65°N and 65°S together with the carbon dioxide concentration, are presented in Fig. 10a. The range of these forcings for the LIG and PI is also shown in Table 1. Fig. 10b illustrates the evolution of simulated global mean temperature by *i*LOVECLIM during LIG compared to PI. The global mean temperature reaches a maximum value of 16.6°C at around 128 kaBP, similar to the peak of carbon dioxide and 1000 years before the summer insolation of 65°N. The temperature difference between 127 kaBP and PI in this work is 0.49°C, which is at the upper end of the range -0.48 to 0.56 °C suggested by the CMIP6/PMIP4 models (Otto-Bliesner et al., 2021). The comparison of the simulated local temperature of *i*LOVECLIM and temperature change proxy which reaches back to 123 kaBP at North GRIP (NGRIP) is shown in Fig. 10c. The simulated local surface temperature at NGRIP peaks at nearly the same time as the global mean temperature (around 128 kaBP). Meanwhile, the proxy-based data shows a similar value around 6500 years later. This could result from the absence of ice sheet and climate interaction in our simulations, as the ice sheet component is not activated. The melting of the ice sheet could possibly delay the increase in temperature. The temperature difference between the LIG and PI at NGRIP in our simulations is 4.2°C, consistent with the range 5.2 ± 2.3°C suggested by Landais et al. (2016). For Antarctica, the comparison of the simulated local surface temperature of *i*LOVECLIM and temperature change proxy at EPICA Dome C (EDC) is presented in Fig. 10d. The change in the simulated temperature shows a good agreement with the proxy-based data regarding timing. However, the warming at EDC during the LIG compared to PI in our work is only 0.59°C while the value suggested by Jouzel et al. (2007) is about 4.5°C. This difference might result from the fixed ice sheet mask and topography in our simulations. It is possible that the West Antarctic Ice Sheet was smaller during

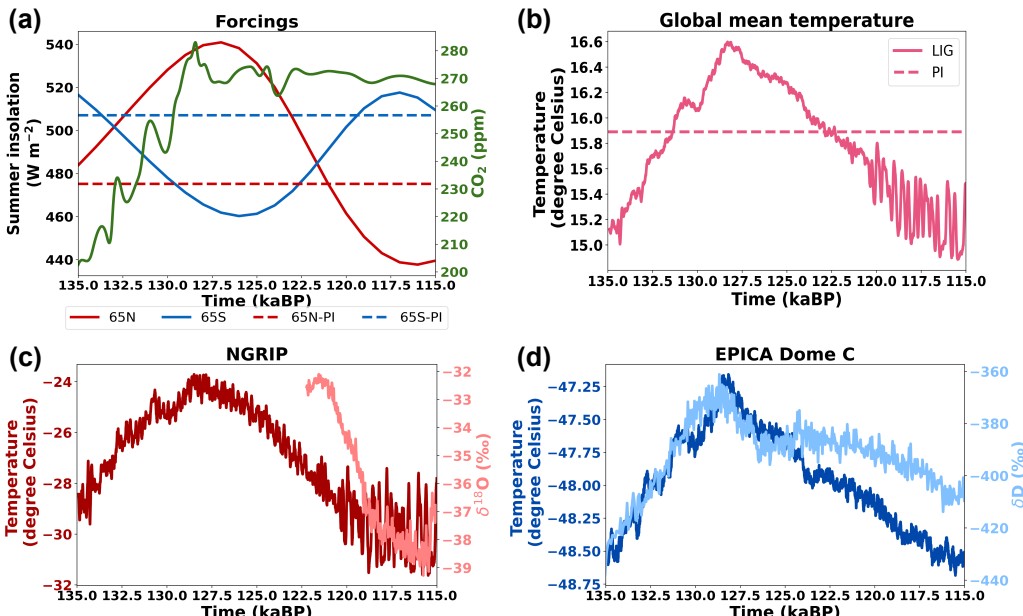

**Figure 10.** (**a**) Temporal variation of external forcings during the LIG: summer insolation of 65°N and 65°S (in W m$^{-2}$) (Berger, 1978) as well as the carbon dioxide concentration (in ppm) (Lüthi et al., 2008). The dashed lines indicates summer insolation of pre-industrial (PI). (**b**) Temporal variation of the 100-year mean of the global mean temperature (in degree Celsius) during the LIG with the value of PI in dashed line. (**c**) The 100-year mean of the simulated local surface temperature (in degree Celsius) and $\delta^{18}O$ (in ‰) (Andersen et al., 2004; Lemieux-Dudon et al., 2010) at North GRIP (NGRIP). (**d**) The 100-year mean of the simulated local surface temperature (in degree Celsius) and $\delta D$ (in ‰) (Jouzel et al., 2007; Lemieux-Dudon et al., 2010) at EPICA Dome C (EDC). The proxy data includes the impact of elevation changes, while our simulations do not.

the LIG, leading to a change in surface elevation and ice extent. This can, in turn, increase the temperature at EDC. This part of warming is not taken into account in our simulations.

The simulated sea ice extent of the Northern and Southern Hemisphere (NH and SH, respectively) during the LIG are shown in Fig. 11a. For both hemispheres, the sea ice extent decreases during the LIG following the temperature changes, reaching the minimum value also around 128 - 127.5 kaBP. The evolution of sea ice extent of the two hemisphere during 127 kaBP in our simulation fall within the range suggested by CMIP6/PMIP4 models (Fig. 4 in Otto-Bliesner et al. (2021)).

### 3.3.2   Surface mass balance evolution during the Last Interglacial

**Greenland**

To study the evolution of SMB during LIG, we present the temporal variation of the annual mean total SMB and its sub-processes simulated by BESSI-*i*LOVECLIM for GrIS in Fig. 12a. The rise of summer insolation in the North and the carbon

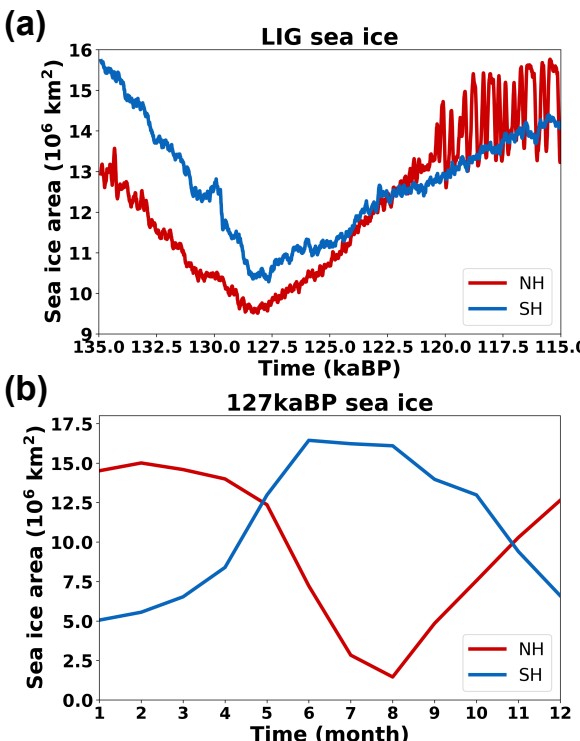

**Figure 11.** Simulated sea ice extent (in $10^6$ km$^2$) for the Northern (NH) and Southern Hemisphere (SH) (**a**) during the LIG and (**b**) during 127 kaBP.

dioxide concentration during the beginning of the LIG (Fig. 10a) induce an increase in the melt rate of Greenland (Fig. 12a). During the same period, the values of runoff are higher than melt's, indicating both rain and melt are not well refreezed due to warm climate (Eq. (A12)). As the insolation drops after 127 kaBP, runoff and melt also decrease. In the same figure, total precipitation is shown to increase slightly during the insolation peak, which is expected as the climate gets warmer. Meanwhile, sublimation/evaporation remains stable throughout the period with a low magnitude as this process is not dominant for GrIS. Similarly, refreezing also remains low for this ice sheet; however, a slight increase during the peak of the LIG is observed in Fig. 12a. The total SMB, in this case, is mainly driven by runoff (melt), strongly decreases during the rise of summer insolation, and then recovers after 127 kaBP. At 128.5 kaBP, the total SMB shrinks to its minimum value (-269.33 Gt yr$^{-1}$), which is around 170% less than the SMB at the beginning of the LIG (372.56 Gt yr$^{-1}$).

Similarly, the annual mean total runoff also increases following the increase of the external forcings in ITM-*i*LOVECLIM (Fig. 12b). However, the magnitude of the runoff is low, leading to the positive total SMB throughout the LIG. This results from low shortwave radiation in the climate forcing (as discussed in Sect. 3.2.1).

By plotting the total SMB differences between the Last Interglacial and the pre-industrial periods simulated by the same model, we investigate the magnitude of SMB variation for both BESSI-*i*LOVECLIM and ITM-*i*LOVECLIM (Fig. 12c). During

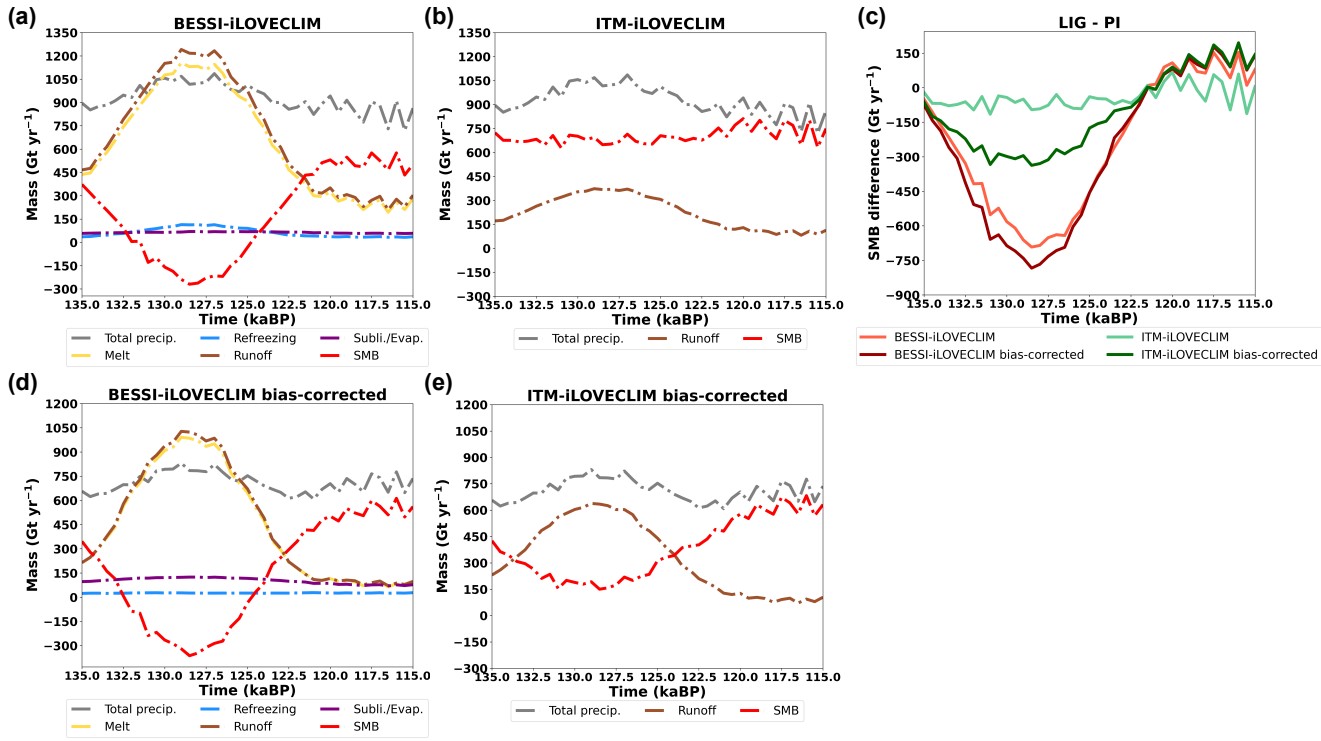

**Figure 12.** Temporal variation of the annual mean total SMB and its elements integrated on the present-day ice sheet extent during the LIG of (**a**) BESSI-*i*LOVECLIM and (**b**) ITM-*i*LOVECLIM (in Gt yr$^{-1}$) for Greenland. (**c**) Annual mean total SMB anomalies between LIG and pre-industrial of different cases.(**d**) and (**e**) Similar like (**a**) and (**b**) but with bias-corrected *i*LOVECLIM

the peak of the LIG, the gap between the two models widens with BESSI-*i*LOVECLIM much lower than ITM-*i*LOVECLIM. The difference between the two models reaches a maximum value of nearly 600 Gt yr$^{-1}$ at 128.5 ka, the same time as the highest global mean temperature. As discussed above, the difference between the two models comes from the runoff simulation, which 350   is also observed in the present-day climate condition (Sect. 3.2.1)

To further investigate the differences between the two SMB schemes, the map of SMB anomalies of BESSI and ITM is shown in Fig. 13. In this figure, we select three different time slices from the LIG simulation: the first (135 kaBP), the peak of the runoff (128.5 kaBP) and the last (115 kaBP) to compare with the pre-industrial results. The pre-industrial annual mean SMB of the two models is quite similar to the present-day value (Fig. 7a). The two models display similar patterns for the first 355   and the last time slices of the LIG. Notably, at the beginning of the LIG, for the two models, positive SMB differences can be seen in the inner part of the ice sheet as there is more precipitation. Meanwhile, SMB rates around the margin are lower than the pre-industrial value since the melting process accelerates due to warmer climate conditions. This SMB trend is enhanced during the peak of deglaciation (128.5 kaBP). In BESSI-*i*LOVECLIM, the magnitude of the negative differences around the margin is very high compared to ITM-*i*LOVECLIM, similar to the present-day climate condition (Fig. 7a). Additionally, BESSI-

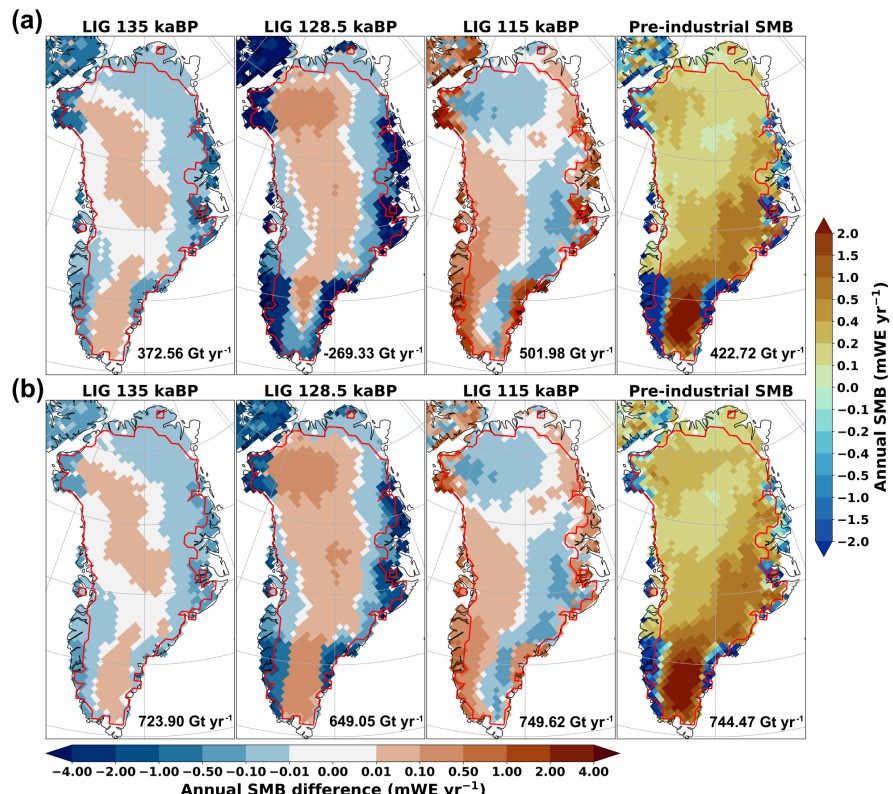

**Figure 13.** Annual mean SMB anomalies (in mWE yr$^{-1}$) between several LIG time slices (135, 128.5 and 115 kaBP) and the pre-industrial simulation of (**a**) BESSI-*i*LOVECLIM and (**b**) ITM-*i*LOVECLIM for Greenland Ice Sheet. The absolute annual SMB value of PI and the total SMB (in Gt yr$^{-1}$) integrated for the present-day ice sheet extent (red line) of each simulation are also included.

*i*LOVECLIM has larger ablation zones with very low SMB than ITM-*i*LOVECLIM, leading to a much lower total SMB rate in BESSI than in ITM (Supplement Fig. S8). Then, at the end of the LIG, both models simulate higher SMB rates around the margins as colder climate accelerates accumulation.

As the climate forcing impacts strongly the simulated SMB, similar runs of BESSI and ITM are carried out with input from the bias-corrected *i*LOVECLIM (Fig. 12 d-e). With this forcing, the simulated total SMB declines for both the SMB models.

Particularly, the minimum total SMB simulated by BESSI-*i*LOVECLIM decreases from -269.33 Gt yr$^{-1}$ to -362.83 Gt yr$^{-1}$ due to a lower total precipitation rate. Similarly, for ITM-*i*LOVECLIM, the minimum total SMB also declines by nearly 500 Gt yr$^{-1}$, from 649.05 Gt yr$^{-1}$ to 150.3 Gt yr$^{-1}$. This reduction results from the lower total precipitation and higher shortwave radiation in bias-corrected *i*LOVECLIM. As the mean total SMB decreases, the SMB anomalies between the LIG and PI of BESSI and ITM also decline (Fig. 12c). At the minimum peak (128 kaBP), LIG-PI anomalies in BESSI-*i*LOVECLIM decreases by over

90 Gt yr$^{-1}$ (from -692 Gt yr$^{-1}$ to -783 Gt yr$^{-1}$). Meanwhile, for ITM, the magnitude of LIG-PI anomalies changes is about 240 Gt yr$^{-1}$, nearly 250% more than before the bias correction (-95 Gt yr$^{-1}$). The results suggest that ITM is more sensitive to the

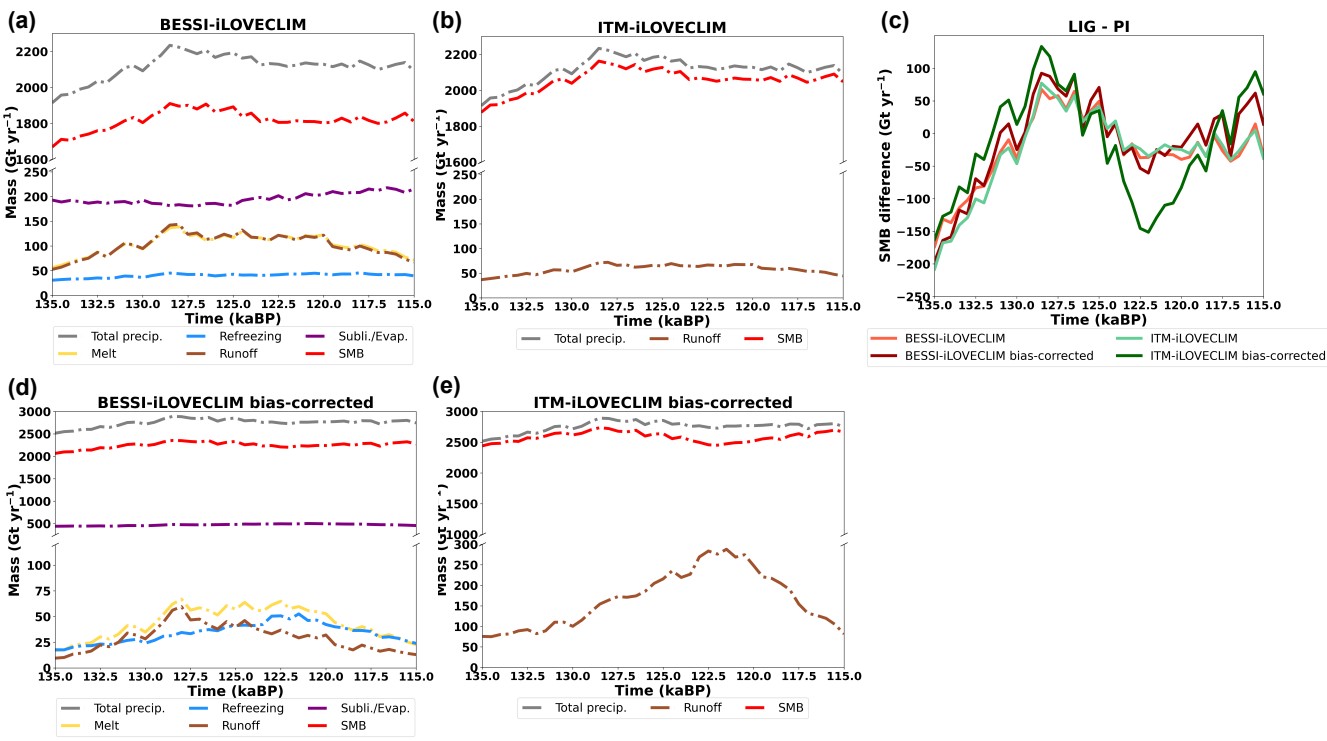

**Figure 14.** Temporal variation of the annual mean total SMB and its elements integrated on the present-day ice sheet extent during LIG of (**a**) BESSI-*i*LOVECLIM and (**b**) ITM-*i*LOVECLIM (in Gt yr$^{-1}$) for Antarctica. (**c**) Annual mean total SMB anomalies between LIG and pre-industrial of different cases.(**d**) and (**e**) Similar like (**a**) and (**b**) but with bias-corrected *i*LOVECLIM

biases in *i*LOVECLIM than BESSI, which is also true for present-day experiments (Fig. 8a). After the bias correction, the simulated SMB patterns are improved for both SMB models with a better shape of ablation zones in GrIS (Supplement Fig. S9).

**Antarctica**

Fig. 14a-b illustrates the temporal variation of the annual mean total SMB and its sub-processes simulated by BESSI-*i*LOVECLIM and ITM-*i*LOVECLIM for AIS. Compared to Greenland, during the same period, the annual mean values of total SMB and its elements fluctuate less in Antarctica for both SMB models. Particularly, in BESSI-*i*LOVECLIM, the total SMB peaks at 128.5 kaBP of 1910.27 Gt yr$^{-1}$, nearly 15% higher than the value of 135 kaBP (1669.16 Gt yr$^{-1}$). This number is very low compared 380 to the -170% differences in GrIS. Also, the magnitude of the simulated annual mean total SMB by BESSI-*i*LOVECLIM during the LIG is quite low for AIS (less than 2000 Gt yr$^{-1}$), which is due to the biases in humidity as discussed in Sect. 3.2.2. During the LIG, even though the insolation at the South Pole decreases (Fig. 10a), AIS still experiences an increase in the melt in BESSI-*i*LOVECLIM (Fig. 12a-b), which is caused by a higher global mean temperature (Fig. 10b). The sublimation is more

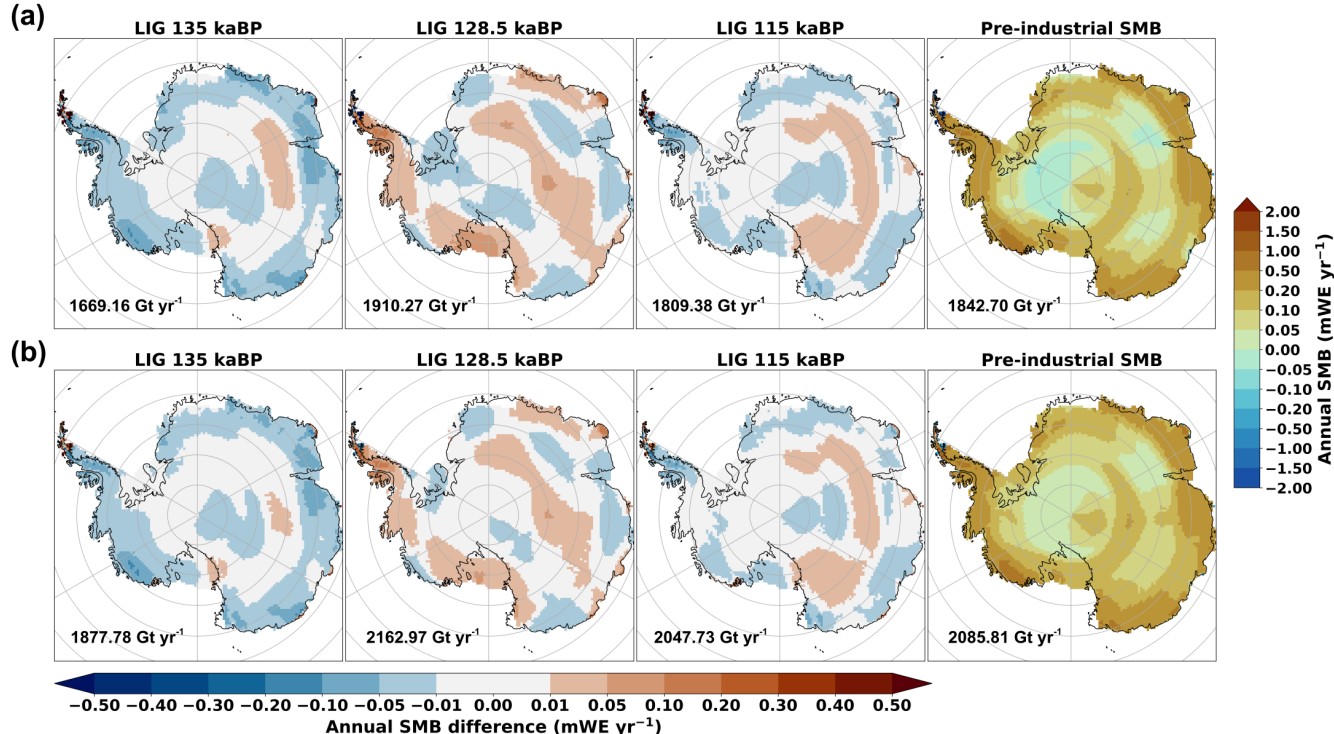

**Figure 15.** Annual mean SMB anomalies (in mWE yr$^{-1}$) between several LIG time slices (135, 128.5 and 115 kaBP) and the pre-industrial simulation of (**a**) BESSI-*i*LOVECLIM and (**b**) ITM-*i*LOVECLIM for Antarctic Ice Sheet. The absolute annual SMB value of PI and the total SMB (in Gt yr$^{-1}$) integrated for the present-day ice sheet extent of each simulation are also included.

dominant in AIS than in GrIS because of a much drier climate. Even though the sublimation is impacted by *i*LOVECLIM

biases, no temporal change of this flux is simulated by the model. This suggests that the influences of the bias in the humidity of *i*LOVECLIM is constant. Due to this and the low value of runoff, for this ice sheet, the variation of the total SMB simulated by BESSI-*i*LOVECLIM follows the pattern of the total precipitation. It slightly increases as the global mean temperature increases since a warmer climate induces more precipitation.

Similarly, the total SMB in ITM-*i*LOVECLIM for AIS is also driven by the total precipitation due to a low rate of runoff

and the absence of sublimation/evaporation processes. The reason for low runoff rates for AIS is the low shortwave radiation simulated by the climate forcing, as discussed in Sect. 3.2.2. Such low runoff rates lead to high total SMB value in ITM-*i*LOVECLIM, which is 2168.97 Gt yr$^{-1}$ at 128.5 ka, 15.5% higher than the value of 135 kaBP.

Fig. 14c indicates that the discrepancies between BESSI and ITM in terms of the SMB anomalies between the LIG and PI are less significant for Antarctica than Greenland (Fig. 12c). As Fig. 14a indicates that sublimation/evaporation is almost

constant during the LIG in BESSI-*i*LOVECLIM, the gap between the two models in Fig. 14c can only be explained by the difference in runoff simulation.

We also investigate the patterns of the annual mean SMB differences between several time slices of LIG and PI in the simulations of BESSI-*i*LOVECLIM and ITM-*i*LOVECLIM (Fig. 15). Similar to GrIS, the pre-industrial annual mean SMB of the two models is also consistent to the present-day results of AIS (Fig. 9a). However, contrary to the GrIS, Fig. 15 suggests not much difference in the SMB value between the LIG and PI as well as between the two models for this ice sheet. This is consistent with Fig. 14c, as the magnitude of the differences is very low compared to that of the absolute SMB value (Supplementary Fig. S10).

For Antarctica, we also investigate the total SMB and its elements with the forcings from the bias-corrected *i*LOVECLIM (Fig. 14d-e). Noticeably, the simulated melt in BESSI-*i*LOVECLIM after bias-corrected reaches its peak at 128.5 kaBP, remains stable for 6000 years before gradually decreasing after 122.5 kaBP. The prolongation of high melt rates is related to the high global mean temperature (Fig. 10b) and the increase of summer insolation of 65°S during this period. The peak of refreezing at 122.5 kaBP indicates that the temperature gets colder, leading to the drop of the melt rate after this time slice (Fig. 14d). Compared to before bias correction, the sublimation/evaporation increases by a factor of two, possibly due to the colder climate, as discussed in Sect. 3.2.2. For ITM-*i*LOVECLIM, the simulated runoff also increases following the increase of the global mean temperature and the summer insolation (Fig. 14e). Around 122.5 - 121.5 kaBP, the runoff rate reaches its peak of nearly 288 Gt yr$^{-1}$, four times the value before bias correction (65 Gt yr$^{-1}$). This results from higher shortwave radiation and higher temperatures in some areas, such as the Wilkes Land (Fig. C2 and Supplement Fig. S11b). Such a big change in the runoff rates leads to the significant difference in LIG-PI anomalies of bias-corrected ITM-*i*LOVECLIM compared to other runs during the period of 122.5 - 121.5 kaBP in Fig. 14c. However, considering the magnitude of the total SMB in AIS, Fig. 14c suggests the LIG-PI anomalies of the runs are not significant both before and after the bias correction.

## 4    Discussion

In this work, we assess the feasibility of replacing a parameterization scheme (ITM) with a physics-based surface energy balance model (BESSI) to provide a more physical SMB approach for the *i*LOVECLIM model framework to simulate the change of ice sheet in the past.

The snow model - BESSI performs well in the calibration/validation with MAR under the present-day climate. Highlighting the model's ability to simulate different climates faithfully, the first-ever simulation for Antarctica (without re-tuning) is in good agreement with MAR, which is more complex and has been intensively used to study this ice sheet. However, the issue related to the strong underestimation of refreezing (Plach et al., 2018; Born et al., 2019) remains (Fig. 5a). Lowering the time step of the model from daily to hourly might solve this problem, as the current model's large time step (daily) neglects the diurnal cycle of temperature (Krebs-Kanzow et al., 2018). On the other hand, the ablation simulated by BESSI is underestimated in extent but mostly overestimated in magnitude. Particularly, the narrow ablation zone in the south-western part of GrIS is underestimated in BESSI-MAR, compared to MAR (Supplement Fig. S1), which is also reported in Fettweis et al. (2020). However, due to the compensation of melt and refreezing, the results of the snow model are in good range with respect to MAR. On the other hand, the parameterization - ITM needs individual tuning for the GrIS and the AIS. Hence, ITM-MAR, with the parameter

$crad$ calibrated for the GrIS, generates an unrealistic runoff rate for the AIS due to the change in the climate condition (e.g., higher shortwave radiation) (Fig. 6a and Supplement Fig. S6a). With a lower $crad$ value, the runoff rates could be reduced to obtain a more suitable total SMB value for Antarctica (Supplement Fig. S12a).

For the paleo study, both BESSI-*i*LOVECLIM and ITM-*i*LOVECLIM simulate the SMB evolution during the LIG following the change of the orbital configuration and carbon dioxide concentration. Despite the influences of the biases in the climate

forcing, the simulated SMB during the 130 - 115 kaBP by BESSI-*i*LOVECLIM of GrIS is in a similar range with the results of MAR and BESSI-MAR from the work of Plach et al. (2018) (Fig. 12). This indicates that BESSI can provide reliable results even when forced by *i*LOVECLIM, a climate forcing with lower resolution than MAR. On the other hand, compared to Sommers et al. (2021), the SMB simulated by BESSI-*i*LOVECLIM both before and after the bias correction is much lower during 127 - 123 kaBP for GrIS. The reason for this is the missing interactive elevation and ice sheet mask. As in this work, we

use the present-day ice sheet topography and extent for all the experiments, leading to the runoff overestimation over parts that had previously melted. Meanwhile, the total SMB simulated by ITM for GrIS remains positive throughout the LIG for both original and bias-corrected *i*LOVECLIM forcings. This suggests that the parameterization is unable to give suitable results without retuning its empirical parameters. As the runoff in ITM is calculated solely by one equation (Eq. (1)), it is easy to have a desired runoff range by tuning its empirical parameters such as $crad$ (Supplement Fig. S12a). Also, the albedo in ITM

is fixed at 0.85, which is the value of ice grid points in *i*LOVECLIM, to give a clean comparison to BESSI. This can also be the reason behind the low runoff simulation in ITM-*i*LOVECLIM during the LIG. A lower albedo value, which means more solar radiation is considered, can increase the simulated runoff rate of ITM (Supplement Fig. S12b). However, using only one albedo value for the whole ice sheet is not realistic. ITM with a range of albedo for different altitudes and locations can provide satisfying results as in Quiquet et al. (2021).

Results of Sect. 3.2 indicates that the quality of the forcings influences both BESSI and ITM. However, the changes in the simulated SMB by BESSI-*i*LOVECLIM before and after bias correction are not as significant as in ITM-*i*LOVECLIM for both ice sheets. The same behaviors of the two models are observed in the results of Sect. 3.3.2. Such sensitivity suggests that ITM needs to be retuned whenever there is a change in the climate forcing in order to obtain desired values. However, this can be problematic for studies focusing on the paleo periods that are not well-documented. Also, a critical limitation of ITM is the

missing sublimation/evaporation processes, which resulted in runoff overestimation. For BESSI, the runoff calculation is more realistic, and more processes are included than just solar radiation and heat. Hence, tuning BESSI is more complicated as it is more physically constrained. Replacing ITM with BESSI to provide SMB to the ice sheet model GRISLI in *i*LOVECLIM framework can produce more physical results. Nonetheless, BESSI requires more input variables than ITM, making it more sensitive to certain biases in *i*LOVECLIM, such as humidity. BESSI is also more computationally expensive (30 years per

minute for the T21 grid) than a parameterization like ITM. However, considering the computational cost of *i*LOVECLIM (500 years per day), the extra cost of having BESSI instead of ITM in the framework is relatively small. In addition, we can be more confident in its response to a change in climate since it explicitly simulates many processes, unlike ITM.

As any climate model, *i*LOVECLIM displays some biases which can be locally dominant (Heinemann et al., 2014). In this work, we investigate the impact of these biases by using a simple delta method to correct the climate of *i*LOVECLIM (see

Appendix C). The results of experiments with *i*LOVECLIM as climate forcings indicate the substantial impacts of these biases on the SMB simulation of both BESSI and ITM. Particularly, the low shortwave radiation provided by *i*LOVECLIM leads to the missing representative of insolation change in ITM, as shown in Sect. 3.3.2. However, transient LIG climate forcings can be obtained with much more favorable computational efficiency thanks to a simple model setup of *i*LOVECLIM. The results of the SMB models are improved with the bias-corrected climate forcings. The results can be further improved with a more

sophisticated bias correction method.

## 5    Conclusions

This work examines the feasibility of replacing the SMB scheme of the Earth system model of intermediate complexity *i*LOVECLIM from a simple parameterization (ITM) by a physics-based surface energy balance model (BESSI) for the purpose of improving the simulation of ice sheet-climate interaction. For this purpose, a comparison between BESSI and ITM stand-

alone is carried out for different climate forcings and climate conditions. Both BESSI and ITM provide acceptable results in the validation in the present-day period by MAR, a state-of-the-art regional climate model that includes a full physical energy mass transfer scheme of the surface for two very different ice sheet climate conditions: GrIS and AIS. For a paleoclimate study, the Last Interglacial period, climate fields simulated by an EMIC called *i*LOVECLIM are used as forcings for both SMB models. *i*LOVECLIM displays a large-scale climate change consistent with the forcings that translate to SMB evolution in agreement

with previous modeling work. Switching from MAR to *i*LOVECLIM highlights the strong influence of the climate forcings on the simulation of the SMB evolution. In particular, *i*LOVECLIM presents important bias that leads to some significant mis-representation of present-day SMB for both GrIS and AIS. These unrealistic climate patterns hamper the performance of both BESSI and ITM, posing the need for bias correction of the climate fields in *i*LOVECLIM. Notably, the comparison between BESSI and ITM during the Last Interglacial suggests a stronger sensitivity of ITM to the biases in the climate forcings. The

current SMB scheme of *i*LOVECLIM needs to be retuned for different climate forcings and study periods, which is not ideal for application in paleo studies. Also, the absence of sublimation/evaporation processes in ITM leads to the overestimation of runoff in order to provide SMB in an acceptable range. The results suggest BESSI can be used to replace ITM as this snow model maintains the low computational cost of *i*LOVECLIM while providing more reliable results without the need to be retuned.

## Appendix A: BESSI model

In the following, we only detail the methodology used for surface energy and mass balance. Full details on the implementation of heat diffusion and snow mass compaction are given in Born et al. (2019).

**Surface energy balance**

The exchange of energy between the surface (the top layer of the model) and the atmosphere results in the change of temperature in this layer ($T_s$), influenced by the net solar flux $Q_{SW}$, the net longwave radiation flux $Q_{LW}$, the sensible heat flux $Q_{SH}$, the latent heat flux $Q_{LH}$, the heat flux from the precipitation $Q_{precip}$ and the melting flux $Q_{melt}$ (when temperature reaches the melting point). This can be expressed as follows

$$c_{ice}m_{top}\frac{\partial T_s}{\partial t}\bigg|_{surface} = Q_{SW} + Q_{LW} + Q_{SH} + Q_{LH}$$
$$+ Q_{precip} + Q_{melt} \tag{A1}$$

in which, $c_i$ is the heat capacity of ice (2110 J kg$^{-1}$K$^{-1}$ at -10°C) and $m_{top}$ is the mass of the top layer in kg m$^{-2}$.

**Table A1.** Table of physical constants and model parameters of the BESSI model.

| Parameter | Symbol | Value | Unit |
|---|---|---|---|
| Albedo of firn | $\alpha_{firn}$ | 0.65 | - |
| Albedo of fresh snow | $\alpha_{freshsnow}$ | 0.82 | - |
| Albedo of ice | $\alpha_{ice}$ | 0.4 | - |
| Coefficient of sensible heat flux | $D_{sh}$ | 15 | W m$^{-2}$ K$^{-1}$ |
| Emissivity of the surface | $\epsilon_{snow}$ | 0.98 | - |
| Density of water | $\rho_{water}$ | 1000 | kg m$^{-3}$ |
| Heat capacity of dry air | $c_{air}$ | 1003 at 0°C | J kg$^{-1}$ K$^{-1}$ |
| Heat capacity of ice | $c_{ice}$ | 2110 at -10°C | J kg$^{-1}$ K$^{-1}$ |
| Heat capacity of water | $c_{water}$ | 4181 at 25°C | J kg$^{-1}$ K$^{-1}$ |
| Latent heat of melting | $L_m$ | 3.34×10$^5$ | J kg$^{-1}$ |
| Latent heat of vaporization | $L_v$ | 2.5×10$^6$ | J kg$^{-1}$ |
| Ratio of latent and sensible heat | $r_{lh/sh}$ | 1.0 | - |
| Stefan-Boltzmann constant | $\sigma$ | 5.670373 × 10$^{-8}$ | W m$^{-2}$ K$^{-4}$ |

The net incoming solar radiation $Q_{SW}$ is calculated from the albedo of the surface ($\alpha_{snow}$ or $\alpha_{ice}$) and the incoming shortwave radiation $F_{SW}$ (Wm$^{-2}$) available from the forcing:

$$Q_{SW} = (1-\alpha)F_{SW} \tag{A2}$$

The albedo of ice $\alpha_{ice}$ is fixed at 0.4 while the albedo of snow $\alpha_{snow}$ is calculated considering the exponential decay with time since the last snowfall event (Oerlemans and Knap, 1998; Zolles and Born, 2021):

$$\alpha_{snow} = \alpha_{firn} + (\alpha_{freshsnow} - \alpha_{firn})exp\left(\frac{-N_{snowfall}}{t*}\right) \tag{A3}$$

in which the albedo of firn $\alpha_{firn}$ is 0.6, the albedo of the fresh snow $\alpha_{freshsnow}$ is 0.82, $N_{snowfall}$ is the number of days since the last snowfall event and $t*$ is the number of days for the fresh snow to reach firn condition. Depending on the temperature of the surface $T_s$, $t*$ is set to 20 days for $T_s < 273.15$ K or 5 days for $T_s = 273.15$ K.

The difference between the upcoming longwave radiation $F_{LW}$ from the atmosphere (read from the input) and the emitted longwave radiation flux is the net longwave radiation $Q_{LW}$:

$$Q_{LW} = F_{LW} - \sigma\epsilon_s T_s^4 \tag{A4}$$

in which, $\sigma$ is the Stefan-Boltzmann constant ($5.670373 \times 10^{-8}$ W m$^{-2}$ K$^{-4}$), $\epsilon_s$ is the emissivity of the snow (0.98).

The turbulent sensible heat flux $Q_{SH}$ equals to the difference between the temperature of the air $T_{air}$ and that of the surface layer $T_s$ multiplied by a coefficient $D_{sh}$ (15 W m$^{-2}$ K$^{-1}$):

$$Q_{SH} = D_{sh}(T_{air} - T_s) \tag{A5}$$

The turbulent latent heat flux $Q_{LH}$ depends on the difference between the water vapor pressure of the air $e_{air}$ and of the surface layer $e_s$, the surface pressure $p_{air}$ from input and a coefficient $D_{lh}$:

$$Q_{LH} = \frac{D_{lh}}{p_{air}}(e_{air} - e_s) \tag{A6}$$

with $D_{lh} = 0.622r_{lh/sh}\frac{D_{sh}}{c_{air}}(L_v + L_m) \tag{A7}$

where $r_{lh/sh}$ is the ratio of the exchange rates between the latent heat and sensible heat (equal to 1.0 in this work), $c_{air}$ is the heat capacity of the air (1003 J kg$^{-1}$ K$^{-1}$ at 0°C) whilst $L_v$ and $L_m$ are latent heat of vaporization and melting, respectively ($2.5\times10^6$ J kg$^{-1}$ and $3.34\times10^5$ J kg$^{-1}$). Details of the turbulent sensible and latent heat fluxes calculation methods are available in Zolles and Born (2021).

Based on the air temperature ($T_{air}$), BESSI classifies total precipitation as snow ($T_{air} \leq 273.15$ K) or rain ($T_{air} > 273.15$ K). When snow/rain falls, the air temperature is transported to the surface. Hence, the equations of heat flux from the snow/rain are:

$$Q_{precip,s} = m_{precip}c_i(T_{air} - T_s) \tag{A8}$$

$$Q_{precip,r} = m_{precip}c_w(T_{air} - 273.15) \tag{A9}$$

where $m_{precip}$ is the mass of precipitation (kg m$^{-2}$ d$^{-1}$) and $c_{water}$ is heat capacity of water (4181 J kg$^{-1}$ K$^{-1}$ at 25°C).

The model uses an implicit scheme, for which the energy fluxes are calculated first, then the energy required to heat the top layer to the melting point. As the temperature of the surface cannot exceed the melting point, the remaining energy is considered as energy available to melt snow/ice $Q_{melt}$ (Eq. (A1)). The main parameters of the model are presented in Table A1.

## Surface mass balance

Surface mass balance SMB is an important element of the ice sheet mass balance, apart from the ice discharge and basal melting. In BESSI, SMB is calculated as the remaining mass of total precipitation from runoff and sublimation/evaporation processes:

$$SMB = m_{precip} - (m_{runoff} + m_{sub}) \tag{A10}$$

In BESSI, the incoming precipitation (rain/snow) accumulates first on the surface (Fig. 1). Generally, the precipitation adds snow mass to the top snow layer ($T_{air} \leq 273.15$ K) or liquid mass to the water content of the surface ($T_{air} > 273.15$ K). As more snow accumulates in the top layer, BESSI generates new snow layers below to prevent the mass of the layer from exceeding the maximum threshold (500 kg m$^{-2}$). The mass of the new layer is set at 300 kg m$^{-2}$, and the old layer keeps the remaining mass, continuing to accumulate snow. Depending on the precipitation and the temperature, up to 15 layers can be formed. When more than one layer exists, the masses of these layers are shifted down to leave space for the new forming layer. In contrast, when $Q_{melt}$ is available, the snow column melts from the top. To prevent the mass of the surface layer from sinking below the minimum threshold (100 kg m$^{-2}$), BESSI merges this layer with the next one. After the merging, the masses of the layers below are shifted up. In case $Q_{melt}$ is enough to melt all the snow layers, ice starts to melt, adding water to the runoff.

The water resulting from melt and rain is retained by the snow column up to 10% of its pore volume. The excess water percolates through the snow column, either refreezing due to low temperatures or leaving the lowest layer as runoff. The energy for refreezing, according to the assumption that the snow and the liquid water inside the snowpack are in thermodynamic equilibrium (Born et al., 2019), is calculated as:

$$Q_{refreezing} = c_i m_s (273.15 - T_{snow}) \tag{A11}$$

in which $T_{snow}$ is the temperature of the snow layer where the process takes place. Refreezing can occur anywhere among the snow layers, unlike melt, which happens only at the top.

The resulting amount of water from processes of rain, melt, and refreezing that leaves the bottom layer is considered as runoff:

$$\frac{\partial m_{runoff}}{\partial t} = m_{rain} + m_{melt} - m_{refreezing} \geq 0 \tag{A12}$$

Sublimation/Evaporation, depending on the humidity of the air, is converted from the turbulent latent heat flux $Q_{LH}$ to mass as:

$$\frac{\partial m_{sub}}{\partial t} = -\frac{Q_{LH}}{L_v + L_m} \tag{A13}$$

Positive values indicate sublimation/evaporation happens, subtracting mass from SMB. On the contrary, deposition/condensation occurs, adding mass to SMB.

## Appendix B:  Evaluation metrics for BESSI-MAR and ITM-MAR with MAR as climate forcing

The goodness-of-fit metrics used to evaluate behaviors of BESSI-MAR and ITM-MAR for the present-day climate condition are presented in the following. The coefficient of determination $R^2$ is calculated as:

$$R^2 = 1 - \frac{\sum_i^n (X_{BESSI_i} - X_{MAR_i})^2}{\sum_i^n (X_{MAR_i} - \overline{X_{MAR}})^2} \tag{B1}$$

in which $(X_{BESSI_i} - X_{MAR_i})$ is the difference between the climatological annual mean value of the same variable of BESSI-MAR and MAR for the grid cell $i$. $\overline{X_{MAR}}$ indicates the spatial mean value of MAR of 43-year-mean result.

The Root Mean Squared Errors (RMSE) is defined as:

$$RMSE = \sqrt{\frac{1}{n} \sum_i^n (X_{BESSI_i} - X_{MAR_i})^2} \tag{B2}$$

Here, $n$ is the total number of the grid points of each ice sheet domain: 7,665 for GrIS and 11,217 for AIS, which is also the case for Eq.B1. The same equations are applied to ITM-MAR.

## Appendix C:  Bias correction procedure for *i*LOVECLIM

To investigate the influence of the climate biases in *i*LOVECLIM on BESSI and ITM behaviors, we use the delta method to correct these biases with ERA5 (Muñoz-Sabater et al., 2021), a reanalysis climate data, as reference.

Input for BESSI includes near-surface temperature, precipitation, surface pressures, humidity, and short-/long-wave radiation in daily time step. For temperature, the bias-corrected data is obtained as follows:

$$T'_{iLC} = T_{iLC} + (\overline{T_{ERA5}} - \overline{T_{iLC}}) \tag{C1}$$

in which, $T'_{iLC}$ is the bias-corrected daily temperature of *i*LOVECLIM, $T_{iLC}$ is the origin daily output of *i*LOVECLIM, $(\overline{T_{ERA5}} - \overline{T_{iLC}})$ is the differences in the daily climatological mean temperature of the period 1979-2021 between ERA5 and *i*LOVECLIM.

For other input variables, the bias correction is carried out as:

$$X'_{iLC} = X_{iLC} \times \frac{\overline{X_{ERA5}}}{\overline{X_{iLC}}} \tag{C2}$$

in which, $X'_{iLC}$ is the bias-corrected daily data of *i*LOVECLIM, $X_{iLC}$ is the origin daily output of *i*LOVECLIM, $\overline{X_{ERA5}}$ and $\overline{X_{iLC}}$ is the daily climatological mean data of the period 1979-2021 correspondent to the reference (ERA5) and *i*LOVECLIM.
In order to avoid extreme value, the ratio $\frac{\overline{X_{ERA5}}}{\overline{X_{iLC}}}$ is limited to be in the range of 0.1 to 10.0. In addition, for relative humidity only, once the bias is corrected, the value is restricted between 0.15 and 1.0 (15-100%) to avoid unrealistic values. These bias correction factors are presented in Fig. C1 for GrIS and Fig. C2 for AIS.

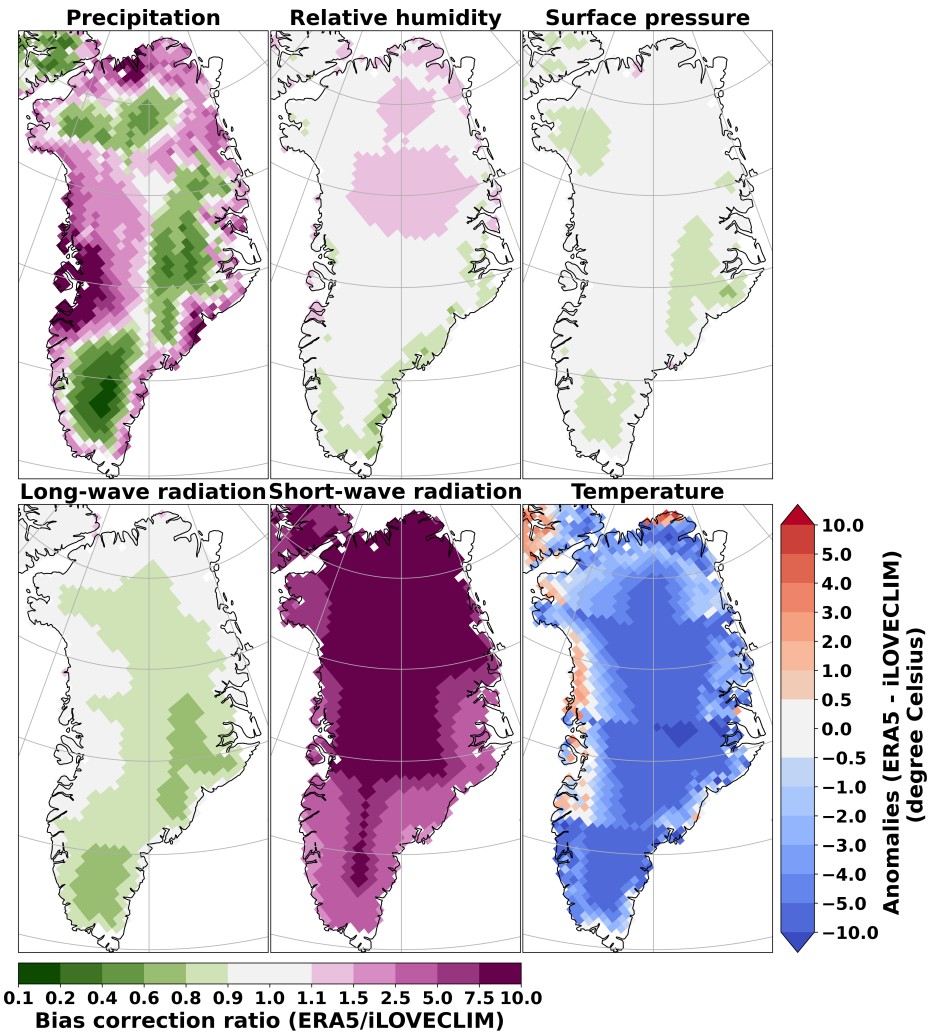

**Figure C1.** Mean values of bias correction factors of *i*LOVECLIM respect to ERA5 for Greenland.

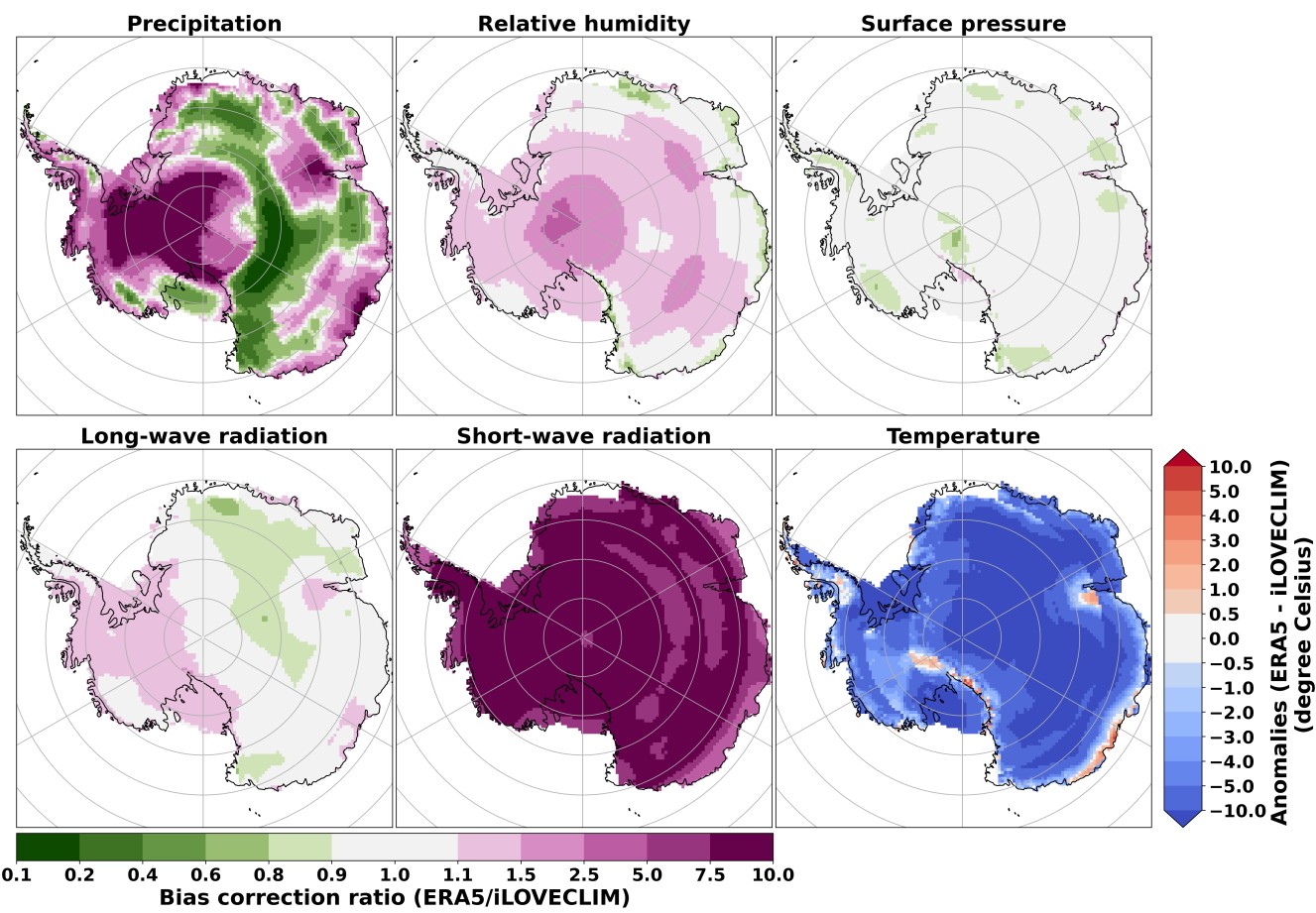

**Figure C2.** Mean values of bias correction factors of *i*LOVECLIM respect to ERA5 for Antarctica.

*Data availability.* Archiving of source data of the figures presented in the main text of the manuscript is underway. Data will be made publicly available upon publication of the manuscript on the Zenodo repository with digital object identifier 10.xxxx/zenodo.xxxxxxx. They are temporarily available for review purposes upon request.

*Code availability.* BESSI version used in this work will be made publicly available upon publication of the manuscript on the Zenodo repository with digital object identifier 10.xxxx/zenodo.xxxxxxx.

*Author contributions.* TKDH and AQ designed the study with contributions from CD and DMR. AB provided the source code of the BESSI model. All authors contributed to the analysis of the results. TKDH performed the simulations and wrote the manuscript with comments from AQ, CD, AB and DMR.

*Competing interests.* The authors declare that they have no conflict of interest.

*Acknowledgements.* Thi-Khanh-Dieu Hoang was supported by the CEA NUMERICS program, which has received funding from the European Union's Horizon 2020 research and innovation program under the Marie Sklodowska-Curie grant agreement No 800945. This work was partially supported by the "PHC AURORA" programme (project number: 51300RA), funded by the French Ministry for Europe and Foreign Affairs, the French Ministry for Higher Education and Research and the Norvegian Council for Research. The authors would like to thank Xavier Fettweis, Charles Amory and Cécile Agosta for providing the data of MAR for GrIS and AIS. We also acknowledge the Institut Pierre Simon Laplace for hosting the iLOVECLIM model code under the LUDUS framework project (http://forge.ipsl.jussieu.fr/ludus, last access: 23 February 2024).

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
