# Peer review of "Using a multi-layer snow model for transient paleo studies: surface mass balance evolution during the Last Interglacial"

_EGUsphere, 2024_

## Referee Comment (RC2)

**1   General**

The manuscript "Improve iLOVECLIM (version 1.1) with a multi-layer snow model: surface mass balance evolution during the Last Interglacial" by Thi-Khanh-Dieu Hoang and Co-authors introduces the BErgen Snow Simulator (BESSI) as a potential coupling interface for ice sheets in the Earth system model iLOVECLIM. BESSI is envisioned to replace the previous

5   iLOVECLIM interface, the less complex, empirical insolation-temperature melt equation (ITM). In contrast to ITM, BESSI is physics based and as such might be applicable for a larger range of background climates (e.g. Greenland Ice Sheet during the last interglacial under orbitally changing insolation; or the Antarctic Ice Sheet with only sporadicly occurring melt and precipitation dominating SMB variability) without the need to retune parameters. The authors first evaluate BESSI for the modern Greenland Ice Sheet using 1979-2021 climate forcing from the regional climate model MAR and comparing it to the

10   respective MAR surface mass balance output. Then BESSI is forced with climate output from a iLOVECLIM simulation of the last interglacial period (LIG, 135000-115000 years before present) and compared to a ITM simulation of the surface mass balance using the same climate forcing.

In my opiniion, the evaluation could go to greater depth. The manuscript is mostly easy to read but the language is sometimes imprecise (anomaly, bias, difference, gap are genorously used quite exchangeable) or sometimes appears to be non-idiomatic.

15   Figures are mostly of good quality but axes labels are often too small. Nonetheless, the implementation of more physics based schemes is surely a desirable improvement of this Earth System Model of intermediate complexity and will be a valuable innovation and I recommend publication after consideration of the following concerns:

**2   Major concerns and general comments**

20   In the Introduction the authors formulate the aim to "answer the question of whether a physics-based scheme can improve the representation of SMB for paleo timescale", but I don't see that the presented results allow to do so. The BESSI surely is more complex and also provides additional information about SMB components, but the results of the interglacial simulation are not necessarily better (possibly only different) than results of the ITM simulation. As a direct comparison to SMB observations for past climates is not possible, I would recommend to add some in depth analysis of the MAR based simulation with respect

25   to the response to qualitatively different constellation (e.g. showcases for early summer versus late summer melt, bare ice in comparison to high accumulation zone, clear sky vs. overcast conditions...). These constellations are probably quite differently represented by BESSI and ITM and might provide insight into the applicability of the individual models for different background conditions.

For the above analysis it would be nice to add a 1979-2021 MAR-ITM simulation which would also allow to directly compare

30   the skill of the two interfaces for present day climate.

At several places the authors claim that ITM is more sensitive to temperature than BESSI - it would be helpful to illustrate this with for example a scatter plot of MELT(BESSI)-MELT(ITM) against temperature and insolation.

35   It is remarkable that ITM, with $c = -25 W m^{-2}$ exhibits a similar sensitivity to interglacial climate change as BESSI with bias correction. With three tuneable parameters it will probably be possible to find ITM parameters which would be in general agreement with BESSI both for preindustrial and last interglacial climate and difference in the behaviour of the two schemes maybe depending on parameter choices in ITM. Also the choice of $c = -25 W m^{-2}$ might represent a first order bias correction of the iLOVECLIM climate. Please discuss.

40

I recommend to evaluate the melt, refreezing and sublimation separately in the figures, as biases are compensating and might be masked by strong precipitation contribution to the signal and e.g. in Fig. S1 the color scale does not resolve differences in sublimation.

Finally I am wondering if this paper would be better placed in a more method-focused journal such as *Geoscientific Model Development*.

**3  Some specific comments**

**title**: "Improve iLOVECLIM"... => "Improved iLOVECLIM" ?

**l.42**: rephrase, maybe: the albedo feedback being absent in the simulation.

**l.43**: maybe better: The first option is to use dedicated snow pack models coupled to RCMs...

**l.61**: maybe: the model's performance => the model's behaviour

**l.77**: please reformulate more carefully as LIG SMB is not known.

**l.78**: also here: I don't see how the advantage can be evaluated. Maybe "evaluate the effect..."

**sect. 2.1.1**: Please highlight changes with respect to the earlier published model version.

**sect. 2.3**: Maybe include a table of all experiments with some climte characteristics (mean JJA temperature and insolation range) and a figure of the topographies used (15km, 40km, T21).

**Fig 2**: Maybe include sublimation,refeezing and melt in the pannels in the middle. Increase font size.

**l.290ff**: a bit clumsy, please rephrase.

**table 2**: Typo: "Greeland", maybe also include bias corrected BESSI.

**l. 305**: the "negtive SMB zone" is somewhat quite comparable to the more extensive ablation zone in MAR. Please discuss.

**l. 324**: "North of AIS" is no good orientation here, maybe Eastern Weddell Sea sector.

**l. 355**: avoid "overestimation" because this would imply that BESSI serves as the reference here- rather use something like "more sensitive than"

**Figure 9, upper panel**: maybe include mean summer temperature for the two ice sheets.

**l. 365**: albedo feedback should be discussed in greater detail.

**l. 374-380**: a bit confusing, please rephrase.

**l. 400**: correct: south-western part of the GrIS

**l. 406**: "cheaper cost" specify the computational cost for orientation (e.g. wall clock time/100 model years) of BESSI-iLOVECLIM and ITM-iLOVECLIM, maybe here, maybe somewhere else...

**l. 415**: more processes do not always increase reliability- additional, poorly constrained feedbacks might actually increase uncertainty...

**l. 420**: insolation is a common forcing for both BESSI and ITM.

**l. 446-447**: check grammar.

**Fig. B2**: Caption does not seem to belong here.

**Fig 1,2,4,5,6,7,8...:** Increase font size.

**References**

Fettweis, X., Hofer, S., Krebs-Kanzow, U., Amory, C., Aoki, T., Berends, C. J., Born, A., Box, J. E., Delhasse, A., Fujita, K., Gierz, P., Goelzer, H., Hanna, E., Hashimoto, A., Huybrechts, P., Kapsch, M.-L., King, M. D., Kittel, C., Lang, C., Langen, P. L., Lenaerts, J. T. M., Liston, G. E., Lohmann, G., Mernild, S. H., Mikolajewicz, U., Modali, K., Mottram, R. H., Niwano, M., Noël, B., Ryan, J. C., Smith, A., Streffing, J., Tedesco, M., van de Berg, W. J., van den Broeke, M., van de Wal, R. S. W., van Kampenhout, L., Wilton, D., Wouters, B., Ziemen, F., and Zolles, T.: GrSMBMIP: intercomparison of the modelled 1980–2012 surface mass balance over the Greenland Ice Sheet, The Cryosphere, 14, 3935–3958, https://doi.org/10.5194/tc-14-3935-2020, https://tc.copernicus.org/articles/14/3935/2020/, 2020.

85

---

## Author Comment (AC1)

**In the following, the reviewer's comments are in blue, and our response is in black.**

**General comments**

The paper by Hoang et al. describes the implementation of the simple surface energy and mass balance model BESSI into the iLOVECLIM Earth system model. First, the performance of BESSI for the simulation of the present-day SMB over Greenland and Antarctica is compared with results of the regional climate model MAR. Then BESSI is driven by climate forcing from iLOVECLIM simulations to show how climate model biases affect the SMB and how the results compare to the insolation-temperature-melt (ITM) method used previously in iLOVECLIM. Finally, transient iLOVECLIM simulations of the last interglacial period are performed and climate fields are used to drive BESSI and estimate the evolution of the SMB and its different components over the time interval from 135 to 115 kyrBP. The results indicate that BESSI and the ITM produce different SMB evolutions for the Greenland ice sheet over this period of time. I have some major comments that should be addressed before this paper can be published in Climate of the Past.

We thank the reviewer for the useful comments. These comments are taken into account in the paper's revised version. Detailed answers to specific comments are provided in the following section.

Here are the main adjustments to the revised manuscript:

- The title of the paper has been changed to make it less technical.
- The description of the BESSI model is shortened in the main text and moved to Appendix A.
- ITM stand-alone replaces the previous version, which runs with the iLOVECLIM time step (4-hourly) and includes the impact of iLOVECLIM temperature biases on the parameter crad of the ITM equation (Quiquet et al., 2024). In the revised version, BESSI is compared to ITM stand-alone, which uses the same input and runs in the same time step. This provides a cleaner comparison and insights into the two models' behavior, as suggested by Reviewer 2.
- As both reviewers suggested, the bias correction of iLOVECLIM is now included in the main text.
- To put more weight on the Last Interglacial (LIG), as both reviewers suggested, we shortened the part of calibration/validation of BESSI and ITM (stand-alone) with only important results. The climate simulated by iLOVECLIM for the LIG is also evaluated based on existing knowledge.

The paper is rather technical, with a large part being dedicated to the description of the BESSI model (which is already described in detail in Born et al. 2018) and model evaluation for the present-day Greenland and Antarctic ice sheets. This part of the paper would actually better fit the scope of a journal like Geoscientific Model Development.

In order to make the manuscript less technical, we have reshaped the new version to focus more on the differences between BESSI and ITM, as well as the effects of using BESSI for paleo studies. We have also changed the paper's title to "Using a multi-layer snow model for transient paleo studies: surface mass balance evolution during the Last Interglacial." The description of the BESSI model has now been moved to an appendix.

However, I think that there is the potential to make the more scientific part of the paper more prominent to make the publication generally more interesting for readers of Climate of the Past. To make the paper more suitable for Climate of the Past, I would suggest a few changes:

- generally shift the focus of the paper towards the simulations of the LIG
  In the revised version, the paper focuses more on the LIG by comparing the climate simulated by iLOVECLIM with proxy and previous work, as well as the results of BESSI and ITM (stand-alone).

- expand the Introduction with some background information about what is known about the climate and ice sheet evolution (in particular Greenland) over the LIG, both from modeling studies and reconstructions (e.g. sea level)
  The revised version of the Introduction now includes climate and ice sheet evolution information from reconstruction, proxy, and modeling studies (CMIP6/PMIP4) during the LIG.
- add a discussion of the implications of the simulated SMB for the Greenland ice sheet evolution over the LIG
  In the revised version, the paper now includes a part discussing the SMB simulated by BESSI and ITM stand-alone during the LIG.
  In the original version of the manuscript, we used the ITM model that is embedded in iLOVECLIM. This version runs with a 4-hourly time step and includes local "crad" modifications based on the temperature bias of iLOVECLIM with respect to ERA-Interim. Details of this modification are presented in Quiquet et al., 2024. Following reviewer 2's suggestion, we now use a stand-alone version of the ITM so that we can use exactly the same inputs for the ITM and for BESSI, with the same time frequency. It makes the comparison of the two models cleaner. In doing so, discussing the two models' behaviors is also more robust. The description of ITM stand-alone is provided in Section 2.1.3.
- possibly move the Appendix B into the main paper, as the sensitivity analysis is scientifically interesting
  We have replaced the old ITM results with new results from ITM stand-alone to provide a better comparison between ITM and BESSI. The bias correction part is now included in the main text of the paper. The sensitivity of ITM with different crad and albedo values is discussed in the Discussion section with Supplement Fig. S12.

BESSI is certainly more physically-based than the original ITM model, but as also acknowledged by the authors, this does not necessarily imply a more realistic representation of the SMB. Nevertheless, at several places in the text the authors make claims like:

- 'The model also captures well the variation of SMB and its components during the LIG.'
- 'For long-term simulations in paleo studies, BESSI has proved to be able to provide reliable data in a short time as it has a physical model setup and is computationally inexpensive. Particularly, in this work, BESSI-iLOVECLIM simulates well the SMB evolution during the LIG, following the change of the orbital configuration and carbon dioxide concentration.'

To justify statements like these, at the very least the simulated SMB should be compared with previous modelling studies (e.g. Sommers et al. 2021 but also some of the previous work by the authors themselves).

BESSI represents the SMB change in a more physical way than the ITM. However, it is true that we cannot firmly validate the model behaviors in the past, given the lack of solid paleo constraints. We have moderated our statements in the new version of the manuscript and more explicitly compared our results with the existing literature (Plach et al., 2018; Sommers et al., 2021).

Simulated temperature (or more generally climate) changes over Greenland during the early phase of the LIG at a time when the GIS was probably similar to its present-day state could be compared with previous modelling studies (e.g. CMIP6 lig127k) and proxy data.

We agree that the simulated LIG climate of iLOVECLIM was not described in the previous version. In the revised version, the paper now includes a part of the climate of iLOVECLIM for both present-day and LIG. In the LIG climate part, the simulated global mean temperature changes with respect to the pre-industrial and the

sea ice extent are compared to the results of CMIP6 models (Otto-Bliesner et al., 2021). The simulated temperature over the ice sheets is compared to proxy data from ice core NGRIP (for Greenland) and EPICA Dome (for Antarctica) (Andersen et al., 2004; Jouzel et al., 2007; Lemieux-Dudon et al., 2010). The results of these comparisons show that the LIG climate simulated by iLOVECLIM is in the range of previous modeling works and in a reasonably good agreement with the proxy data.

Finally, the English usage in this manuscript must be substantial improved to make the text easier to follow.

We have fixed all the errors the reviewer mentioned. However, as the paper's content has changed a lot, some of the sentences have been removed.

**Minor comments**

L. 1: 'retreat-advanced' -> 'retreat-advance'

Fixed

L. 8: 'an' -> 'the'

Fixed

L. 11: ITM not defined

We are sorry about this. ITM is now correctly defined before using an abbreviation.

L. 16: MaBP is not really a standard abbreviation, possibly use Myr BP or Ma instead?

Ma is now used.

L. 16: I wouldn't call glacial cycles' events'

Corrected.

L. 36-39: maybe worth mentioning that some EMICs are using more sophisticated SMB schemes (e.g. Calov et al. 2005 and Willeit et al. 2024)

Thanks for pointing this out. These models are now included in the Introduction.

L. 44: 'physically key' -> 'key physical'

Fixed.

L. 57: 'a' -> 'the'

Corrected.

L. 66-67: repetition of 'annual global mean temperature'

Fixed.

L. 67-68: the cited references do not support the statement that the LIG was globally 2°C warmer than the pre-industrial. Actually, I believe there is an agreement now that the global temperature change was small, with models indicating no significant change (Otto-Bliesner et al., 2021).

This part has been rewritten following this comment.

The estimation of the global mean temperature change during the LIG with respect to the pre-industrial ranges from almost no change (Capron et al., 2014; Hoffman et al., 2017; Otto-Bliesner et al., 2021) to a 1 to 2°C warming (Turney and Jones, 2010; McKay et al., 2011; Fischer et al., 2018). A warming in the high-latitude regions is nonetheless reported by both proxy data and model outputs

L. 68: retreat of glaciers has certainly little to do with global mean temperature changes

The text has been corrected.

L. 127: 'air temperature is transported to the surface': I would rather say that it is assumed that the temperature of the snow/rain corresponds to the air temperature, right? Considering that precipitation is formed at higher altitudes above the ground where it is usually colder, is this a good approximation? Maybe also provide some details on how large this flux is under typical conditions.

We agree with the reviewer. When falling, snow/rain brings energy to the top layer of the snow model due to the temperature differences. The temperature of snow/rain is assumed to be equal to the air temperature, which is a common approximation (e.g., Greulle and Genthon, 2004; Bougamont et al., 2005; Vizcaíno et al., 2010). This assumption neglects some specific cases, such as the high altitude precipitation, as the reviewer mentioned. However, this assumption is acceptable because BESSI's goal is to be compatible with the low-resolution Earth System Models. Sensible heat from rain is reported to be around 2% of the total energy (Hay and Fitzharris, 1988). Hence, it can be neglected in the energy calculation (Greulle and Genthon, 2004; Krapp et al., 2017; Willeit et al., 2024).

L. 129-130: please specify the units for precip, particularly because it seems to be expressed in non-SI units of m(water equivalent)/s.

The unit for precipitation is $kg/m^2/day$. This information is now provided in the main text. We have also rewritten this part to make it easier to follow.

As the rain falls when the snow condition is not satisfied ($T_{air} > 273.15$ K), BESSI uses the freezing temperature of the water, similar to Vizcaíno et al., 2010. This assumption is acceptable for a large-time step of BESSI (daily).

L. 140-148: If I understand this correctly, as long as the mass of the top layer is below 500 kg/m2, BESSI includes only a single layer. Why is that? Considering that for typical snow densities of ~350 kg/m3 this means that a single layer can be ~1.5 meter thick, this implies that the model is not really resolving vertical gradients and processes in the snow pack.

The splitting mass threshold of 500kg/m2 is to be sure that the mass after the split of the two layers (new and old) is appropriate considering the value of the mass after the splitting. After the split, the new layer is 300 kg/m2, around 1m thick. The old layer now has 200 kg/m2, which is enough so the merging process does not happen soon after splitting and, at the same time, gives enough room for the new accumulation.

We agree that this results in a relatively thick layer at the top, which may not be ideal but is coherent with the time step of one day. Thinner layers would require a much shorter time step, which contradicts one of the primary design goals of BESSI. The model's improvement to have thinner layers is currently under development.

L. 158: Eq. 12 doesn't consider changes in the liquid water/snow storage in the layers, why?

The equation did consider the change of the liquid water/snow storage among the layers. The change in liquid water/snow is considered as follows: the snow melts from the top, and the resulting water percolates through the snow column. The water that reaches the bottom layer is then treated as runoff. The water inside the snow column can be frozen again, adding mass to the snowpack. When the whole snow column melts, the refreezing is zero, and all the water from rain and melt is runoff.

Fixed.

L. 286: And how do you deal with the grid cells that are not covered by the iLOVECLIM land domain?

These grid cells are masked out in the simulation. To illustrate the mask used in the different experiments, we now include the topography map for each simulation in the models' description (Fig. 2 and 3).

L. 291: 'climate major pattern' -> 'major climate patterns'

Fixed.

L. 327-328: ITM is equally unrealistic, as it completely ignores the sublimation

The results of this part have changed.

However, the revised manuscript mentions and discusses the missing sublimation/evaporation processes in ITM.

L. 339-340: sentence unclear

As the results have changed, this sentence is removed.

L. 398: remove 'possibly'

Done

Why not integrate Fig. S1 into Fig. 2?

In the revised version, the SMB differences of BESSI-MAR and ITM-MAR are included with absolute SMB from MAR (Fig.4 and 6). The absolute SMB results are in the Supplement (Fig. S1).

Fig. 6: Maybe worth specifying that ITM is also driven by iLOVECLIM climate output?

We apologize for the confusion. ITM forced by iLOVECLIM is now denoted as ITM-iLOVECLIM.

Fig. 6, 8, 11, 12: The colormap for the SMB is not color-blind-friendly.

The colormap for the SMB is fixed, using the colormap recommended by the publisher (Crameri, 2018).

Fig. 12: wrong units in caption

Fixed (now is Fig.15). The captions of all the figures are double-checked.

Table 2: The Melt from the ITM should rather be compared to the runoff, as refreezing is implicitly accounted for in the ITM model.

In the revised version of the paper, melt from ITM stand-alone is considered as runoff.

**References**

Otto-Bliesner, B. L., Brady, E. C., Zhao, A., Brierley, C. M., Axford, Y., Capron, E., Govin, A., Hoffman, J. S., Isaacs, E., Kageyama, M., Scussolini, P., Tzedakis, P. C., Williams, C. J. R., Wolff, E., Abe-Ouchi, A., Braconnot, P., Ramos Buarque, S., Cao, J., De Vernal, A., Vittoria Guarino, M., Guo, C., Legrande, A. N., Lohmann, G., Meissner, K. J., Menviel, L., Morozova, P. A., Nisancioglu, K. H., O'Ishi, R., Mélia, D. S. Y., Shi, X., Sicard, M., Sime, L., Stepanek, C., Tomas, R., Volodin, E., Yeung, N. K. H., Zhang, Q., Zhang, Z., and Zheng, W.: Large-scale features of Last Interglacial climate: Results from evaluating the lig127k simulations for the Coupled Model Intercomparison Project (CMIP6)-Paleoclimate Modeling Intercomparison Project (PMIP4), Clim. Past, 17, 63–94, https://doi.org/10.5194/cp-17-63-2021, 2021.

Calov, R., Ganopolski, A., Claussen, M., Petoukhov, V., and Greve, R.: Transient simulation of the last glacial inception. Part I: glacial inception as a bifurcation in the climate system, Clim. Dyn., 24, 545–561, https://doi.org/10.1007/s00382-005-0007-6, 2005.

Willeit, M., Calov, R., Talento, S., Greve, R., Bernales, J., Klemann, V., Bagge, M., and Ganopolski, A.: Glacial inception through rapid ice area increase driven by albedo and vegetation feedbacks, Clim. Past, 20, 597–623, https://doi.org/10.5194/cp-20-597-2024, 2024.

Sommers, A. N., Otto-Bliesner, B. L., Lipscomb, W. H., Lofverstrom, M., Shafer, S. L., Bartlein, P. J., Brady, E. C., Kluzek, E., Leguy, G., Thayer-Calder, K., and Tomas, R. A.: Retreat and Regrowth of the Greenland Ice Sheet During the Last Interglacial as Simulated by the CESM2-CISM2 Coupled Climate–Ice Sheet Model, Paleoceanogr. Paleoclimatology, 36, 1–19, https://doi.org/10.1029/2021PA004272, 2021.

**References**

Andersen, K.K., Azuma, N., Barnola, J.-M., Bigler, M., Biscaye, P., Caillon, N., Chappellaz, J., Clausen, H.B., Dahl-Jensen, D., Fischer, H., Flückiger, J., Fritzsche, D., Fujii, Y., Goto-Azuma, K., Grønvold, K., Gundestrup, N.S., Hansson, M., Huber, C., Hvidberg, C.S., Johnsen, S.J., Jonsell, U., Jouzel, J., Kipfstuhl, S., Landais, A., Leuenberger, M., Lorrain, R., Masson-Delmotte, V., Miller, H., Motoyama, H., Narita, H., Popp, T., Rasmussen, S.O., Raynaud, D., Rothlisberger, R., Ruth, U., Samyn, D., Schwander, J., Shoji, H., Siggard-Andersen, M.-L., Steffensen, J.P., Stocker, T., Sveinbjörnsdóttir, A.E., Svensson, A., Takata, M., Tison, J.-L., Thorsteinsson, Th., Watanabe, O., Wilhelms, F., White, J.W.C., North Greenland Ice Core Project members, 2004. High-resolution record of Northern Hemisphere climate extending into the last interglacial period. Nature 431, 147–151. https://doi.org/10.1038/nature02805

Bougamont, M., Bamber, J.L., Greuell, W., 2005. A surface mass balance model for the Greenland Ice Sheet. Journal of Geophysical Research: Earth Surface 110. https://doi.org/10.1029/2005JF000348

Capron, E., Govin, A., Stone, E.J., Masson-Delmotte, V., Mulitza, S., Otto-Bliesner, B., Rasmussen, T.L., Sime, L.C., Waelbroeck, C., Wolff, E.W., 2014. Temporal and spatial structure of multi-millennial temperature changes at high latitudes during the Last Interglacial. Quaternary Science Reviews 103, 116–133. https://doi.org/10.1016/j.quascirev.2014.08.018

Crameri, F., 2018. Geodynamic diagnostics, scientific visualisation and StagLab 3.0. Geoscientific Model Development 11, 2541–2562. https://doi.org/10.5194/gmd-11-2541-2018

Fischer, H., Meissner, K.J., Mix, A.C., Abram, N.J., Austermann, J., Brovkin, V., Capron, E., Colombaroli, D., Daniau, A.-L., Dyez, K.A., Felis, T., Finkelstein, S.A., Jaccard, S.L., McClymont, E.L., Rovere, A., Sutter, J., Wolff, E.W., Affolter, S., Bakker, P., Ballesteros-Cánovas, J.A., Barbante, C., Caley, T., Carlson, A.E., Churakova (Sidorova), O., Cortese, G., Cumming, B.F., Davis, B.A.S., de Vernal, A., Emile-Geay, J., Fritz, S.C., Gierz, P., Gottschalk, J., Holloway, M.D., Joos, F., Kucera, M., Loutre, M.-F., Lunt, D.J., Marcisz, K., Marlon, J.R., Martinez, P., Masson-Delmotte, V., Nehrbass-Ahles, C., Otto-Bliesner, B.L., Raible, C.C., Risebrobakken, B., Sánchez Goñi, M.F., Arrigo, J.S., Sarnthein, M., Sjolte, J., Stocker, T.F., Velasquez Alvárez, P.A., Tinner, W., Valdes, P.J., Vogel, H., Wanner, H., Yan, Q., Yu, Z., Ziegler, M., Zhou, L., 2018. Palaeoclimate constraints on the impact of 2 °C anthropogenic warming and beyond. Nature Geosci 11, 474–485. https://doi.org/10.1038/s41561-018-0146-0

Greuell, W., Genthon, C., 2004. Modelling land-ice surface mass balance, in: Payne, A.J., Bamber, J.L. (Eds.), Mass Balance of the Cryosphere: Observations and Modelling of Contemporary and Future Changes. Cambridge University Press, Cambridge, pp. 117–168. https://doi.org/10.1017/CBO9780511535659.007

Hay, J.E., Fitzharris, B.B., 1988. A Comparison of the Energy-Balance and Bulk-aerodynamic Approaches for Estimating Glacier Melt. Journal of Glaciology 34, 145–153. https://doi.org/10.3189/S0022143000032172

Hoffman, J.S., Clark, P.U., Parnell, A.C., He, F., 2017. Regional and global sea-surface temperatures during the last interglaciation. Science 355, 276–279. https://doi.org/10.1126/science.aai8464

Jouzel, J., Masson-Delmotte, V., Cattani, O., Dreyfus, G., Falourd, S., Hoffmann, G., Minster, B., Nouet, J., Barnola, J.M., Chappellaz, J., Fischer, H., Gallet, J.C., Johnsen, S., Leuenberger, M., Loulergue, L., Luethi, D., Oerter, H., Parrenin, F., Raisbeck, G., Raynaud, D., Schilt, A., Schwander, J., Selmo, E., Souchez, R., Spahni, R., Stauffer, B., Steffensen, J.P., Stenni, B., Stocker, T.F., Tison, J.L., Werner, M., Wolff, E.W., 2007. Orbital and Millennial Antarctic Climate Variability over the Past 800,000 Years. Science 317, 793–796. https://doi.org/10.1126/science.1141038

Krapp, M., Robinson, A., Ganopolski, A., 2017. SEMIC: an efficient surface energy and mass balance model applied to the Greenland ice sheet. The Cryosphere 11, 1519–1535. https://doi.org/10.5194/tc-11-1519-2017

Lemieux-Dudon, B., Blayo, E., Petit, J.-R., Waelbroeck, C., Svensson, A., Ritz, C., Barnola, J.-M., Narcisi, B.M., Parrenin, F., 2010. Consistent dating for Antarctic and Greenland ice cores. Quaternary Science Reviews, Climate of the Last Million Years: New Insights from EPICA and Other Records 29, 8–20. https://doi.org/10.1016/j.quascirev.2009.11.010

McKay, N.P., Overpeck, J.T., Otto-Bliesner, B.L., 2011. The role of ocean thermal expansion in Last Interglacial sea level rise. Geophysical Research Letters 38. https://doi.org/10.1029/2011GL048280

Otto-Bliesner, B.L., Brady, E.C., Zhao, A., Brierley, C.M., Axford, Y., Capron, E., Govin, A., Hoffman, J.S., Isaacs, E., Kageyama, M., Scussolini, P., Tzedakis, P.C., Williams, C.J.R., Wolff, E., Abe-Ouchi, A., Braconnot, P., Ramos Buarque, S., Cao, J., de Vernal, A., Guarino, M.V., Guo, C., LeGrande, A.N., Lohmann, G., Meissner, K.J., Menviel, L., Morozova, P.A., Nisancioglu, K.H., O'ishi, R., Salas y Mélia, D., Shi, X., Sicard, M., Sime, L., Stepanek, C., Tomas, R., Volodin, E., Yeung, N.K.H., Zhang, Q., Zhang, Z., Zheng, W., 2021. Large-scale features of Last Interglacial climate: results from evaluating the *lig127k* simulations for the Coupled Model Intercomparison Project (CMIP6)–Paleoclimate Modeling Intercomparison Project (PMIP4). Climate of the Past 17, 63–94. https://doi.org/10.5194/cp-17-63-2021

Plach, A., Nisancioglu, K.H., Le clec'h, S., Born, A., Langebroek, P.M., Guo, C., Imhof, M., Stocker, T.F., 2018. Eemian Greenland SMB strongly sensitive to model choice. Climate of the Past 14, 1463–1485. https://doi.org/10.5194/cp-14-1463-2018

Quiquet, A., Roche, D.M., 2024. Investigating similarities and differences of the penultimate and last glacial terminations with a coupled ice sheet–climate model. Climate of the Past 20, 1365–1385. https://doi.org/10.5194/cp-20-1365-2024

Sommers, A.N., Otto-Bliesner, B.L., Lipscomb, W.H., Lofverstrom, M., Shafer, S.L., Bartlein, P.J., Brady, E.C., Kluzek, E., Leguy, G., Thayer-Calder, K., Tomas, R.A., 2021. Retreat and Regrowth of the Greenland Ice Sheet During the Last Interglacial as Simulated by the CESM2-CISM2 Coupled Climate–Ice Sheet Model. Paleoceanography and Paleoclimatology 36, e2021PA004272. https://doi.org/10.1029/2021PA004272

Turney, C.S.M., Jones, R.T., 2010. Does the Agulhas Current amplify global temperatures during super-interglacials? Journal of Quaternary Science 25, 839–843. https://doi.org/10.1002/jqs.1423

Vizcaíno, M., Mikolajewicz, U., Jungclaus, J., Schurgers, G., 2010. Climate modification by future ice sheet changes and consequences for ice sheet mass balance. Clim Dyn 34, 301–324. https://doi.org/10.1007/s00382-009-0591-y

Willeit, M., Calov, R., Talento, S., Greve, R., Bernales, J., Klemann, V., Bagge, M., Ganopolski, A., 2024. Glacial inception through rapid ice area increase driven by albedo and vegetation feedbacks. Climate of the Past 20, 597–623. https://doi.org/10.5194/cp-20-597-2024

---

## Author Comment (AC2)

**In the following, the reviewer's comments are in blue, and our response is in black**

**1 General**

The manuscript "Improve iLOVECLIM (version 1.1) with a multi-layer snow model: surface mass balance evolution during the Last Interglacial" by Thi-Khanh-Dieu Hoang and Co-authors introduces the BErgen Snow Simulator (BESSI) as a potential coupling interface for ice sheets in the Earth system model iLOVECLIM. BESSI is envisioned to replace the previous iLOVECLIM interface, the less complex, empirical insolation-temperature melt equation (ITM). In contrast to ITM, BESSI is physics based and as such might be applicable for a larger range of background climates (e.g. Greenland Ice Sheet during the last interglacial under orbitally changing insolation; or the Antarctic Ice Sheet with only sporadicly occurring melt and precipitation dominating SMB variability) without the need to retune parameters. The authors first evaluate BESSI for the modern Greenland Ice Sheet using 1979-2021 climate forcing from the regional climate model MAR and comparing it to the respective MAR surface mass balance output. Then BESSI is forced with climate output from a iLOVECLIM simulation of the last interglacial period (LIG, 135000-115000 years before present) and compared to a ITM simulation of the surface mass balance using the same climate forcing.

In my opinion, the evaluation could go to greater depth. The manuscript is mostly easy to read but the language is sometimes imprecise (anomaly, bias, difference, gap are genorously used quite exchangeable) or sometimes appears to be non-idiomatic. Figures are mostly of good quality but axes labels are often too small. Nonetheless, the implementation of more physics based schemes is surely a desirable improvement of this Earth System Model of intermediate complexity and will be a valuable innovation and I recommend publication after consideration of the following concerns:

We thank the reviewer for the useful comments. These comments are taken into account in the paper's revised version. Detailed answers to specific comments are provided in the following section.

Here are the main adjustments to the revised manuscript:

- The title of the paper has been changed to make it less technical.
- The description of the BESSI model is shortened in the main text and moved to Appendix A.
- ITM stand-alone replaces the previous version, which runs with the iLOVECLIM time step (4-hourly) and includes the impact of iLOVECLIM temperature biases on the parameter crad in the ITM equation (Quiquet et al., 2024). In the revised version, BESSI is compared to ITM stand-alone, which uses the same input and runs in the same time step. This provides a cleaner comparison and insights into the two models' behavior, as suggested by the reviewer.
- As both reviewers suggested, the bias correction of iLOVECLIM is now included in the main text.
- To put more weight on the Last Interglacial (LIG), as both reviewers suggested, we shortened the part of calibration/validation of BESSI and ITM (stand-alone) with only important results. The climate simulated by iLOVECLIM for the LIG is also evaluated based on existing knowledge.
- All the figures are fixed with the appropriate colorbar, bigger font size, and correct captions.

**2 Major concerns and general comments**

In the Introduction the authors formulate the aim to "answer the question of whether a physics-based scheme can improve the representation of SMB for paleo timescale", but I don't see that the presented results allow to do so. The BESSI surely is more complex and also provides additional information about SMB components, but the results of the interglacial simulation are not necessarily better (possibly only different) than results of

the ITM simulation. As a direct comparison to SMB observations for past climates is not possible, I would recommend to add some in depth analysis of the MAR based simulation with respect to the response to qualitatively different constellation (e.g. showcases for early summer versus late summer melt, bare ice in comparison to high accumulation zone, clear sky vs. overcast conditions...). These constellations are probably quite differently represented by BESSI and ITM and might provide insight into the applicability of the individual models for different background conditions.

For the above analysis it would be nice to add a 1979-2021 MAR-ITM simulation which would also allow to directly compare the skill of the two interfaces for present day climate.

It is true that we cannot validate BESSI and ITM for the paleo period like the LIG. As the reviewer suggested, we compare BESSI to ITM stand-alone in the revised manuscript. In the previous version of the manuscript, we used the ITM embedded in iLOVECLIM, which is not directly comparable to BESSI as it runs in 4-hourly timestep as iLOVECLIM and contains some modifications related to the temperature biases in iLOVECLIM and albedo value as mentioned in Sect 2.1.3.

For the stand-alone version, ITM is forced by a similar input as BESSI and runs in daily timestep. In iLOVECLIM, the albedo of ice grid cells is set to be 0.85. Hence, ITM stand-alone also uses the same value for albedo in the equation. The crad value is obtained by tuning to achieve the best match for total surface mass balance (SMB), taken to be equivalent to the lowest RMSE compared to MAR (Sect. 2.3).

To investigate the behaviors of BESSI and ITM for the present-day condition with MAR as forcing, we select 2 points in the accumulation zone and 2 points in the ablation zone. Point 1 is in an accumulation zone with a cold and dry climate. In a similar zone, point 2 has mild and wet conditions. Points 3 and 4 are similar to points 1 and 2 but located in the ablation zone. By selecting points from different climate conditions (humidity and temperature) for bare ice and accumulation zones, we compare the behaviors of MAR, BESSI and ITM. The location of the points is provided in Fig. RA1a.

For each point, we plot surface mass balance - SMB, runoff - RU and sublimation - SU of MAR, BESSI-MAR, and ITM-MAR climatological mean daily variation. Following the suggestion of reviewer 1, melt computed by ITM is considered as runoff. In general, compared to MAR and BESSI, ITM tends to simulate runoff earlier due to the sensitivity to the temperature while missing the explicit refreezing calculation.

Differences in each process of the three models are provided in annual mean maps in the Supplement (Fig. S2 and S3 for Greenland and S6 for Antarctica).

[Figure]

**Figure RA1. (a)** The location of four selected points on the Greenland Ice Sheet. The climatological variation of inputs (temperature and humidity), SMB, RU and SU for **(b)** point 1, **(c)** point 2, **(d)** point 3 and **(e)** point 4.

[Figure]

**Figure RA2.** Mean runoff differences between BESSI and ITM forced by iLOVECLIM of Greenland Ice Sheet during the LIG respect to the mean temperature anomalies (compared to 135 kaBP) and the summer insolation at 65N **(a)** before and **(b)** after the bias-correction.

At several places the authors claim that ITM is more sensitive to temperature than BESSI - it would be helpful to illustrate this with for example a scatter plot of MELT(BESSI)-MELT(ITM) against temperature and insolation.

In the revised version, the paper includes more results and a discussion on the sensitivity of ITM to the input data (temperature and short-wave radiation). Since the melt in ITM is treated as Runoff to compare to Runoff of BESSI, as Reviewer 1 suggested, the heat map of Runoff(BESSI)-Runoff(ITM) against local temperature anomalies (compared to 135 kaBP) and summer insolation of Greenland Ice Sheet during the LIG are presented in Fig. RA2. We show a heatmap instead of a scatter plot since we only have discontinuous values for the insolation (41 time slices). Fig. RA2 displays the annual mean runoff anomalies between BESSI and ITM spatially averaged for a given temperature range and a given summer insolation. In general, the figure indicates that the role of temperature is more significant than insolation since the horizontal gradients (impact of temperature change) are larger than the vertical gradients (effect of insolation change). However, it is hard to draw any firm conclusion from this figure since the temperature change integrates all the other changes, including insolation change.

The comparison of the simulated runoff rate by BESSI and ITM is also shown in Fig. 12 for Greenland and Fig. 14 for Antarctica in the revised manuscript. From these figures, the change of runoff in ITM due to the insolation and temperature is more significant. In addition, these figures indicate that ITM is sensitive to its inputs and needs to be retuned when the climate conditions change.

It is remarkable that ITM, with $c = -25 Wm^{-2}$ exhibits a similar sensitivity to interglacial climate change as BESSI with bias correction. With three tuneable parameters it will probably be possible to find ITM parameters which would be in general agreement with BESSI both for preindustrial and last interglacial climate and difference in the behaviour of the two schemes maybe depending on parameter choices in ITM. Also the choice of $c = -25 Wm^{-2}$ might represent a first order bias correction of the iLOVECLIM climate. Please discuss.

ITM inside iLOVECLIM is partly bias corrected by changing the crad with respect to the bias in the temperature (Quiquet et al., 2024). For example, when the temperature difference between iLOVECLIM and ERA-Interim is +10°C, the crad value is double, which means less runoff to account for this warm bias.

In the revised version, we use ITM stand-alone with one crad value for the whole ice sheet. In the new results, we can see that even when both BESSI and ITM are calibrated to behave similarly to MAR for Greenland, their behaviors start to be different when the climate conditions or the forcings change. For Antarctica, unlike BESSI, without retuning, ITM simulates unrealistic high melt with MAR as the forcing.

I recommend to evaluate the melt, refreezing and sublimation separately in the figures, as biases are compensating and might be masked by strong precipitation contribution to the signal and e.g. in Fig. S1 the color scale does not resolve differences in sublimation.

Plots of the differences in simulated albedo, melt, refreezing, runoff, and sublimation of BESSI-MAR, ITM-MAR, and MAR for Greenland and Antarctica are now in the Supplement.

Finally I am wondering if this paper would be better placed in a more method-focused journal such as Geoscientific Model Development.

We agree that the title and the description of the BESSI model make the paper seem technical. To fix this, we have reshaped the manuscript to focus more on the differences between BESSI and ITM as well as the effects of using BESSI for the paleo studies. We have also changed the paper's title to "Using a multi-layer snow model for transient paleo studies: surface mass balance evolution during the Last Interglacial." The BESSI model's description has now been moved to the Appendix to make the paper less technical.

**3 Some specific comments**

We have fixed all the errors the reviewer mentioned. However, as the paper's content has changed a lot, some of the sentences have been removed.

title: "Improve iLOVECLIM"... => "Improved iLOVECLIM"?

We have changed the title.

l.42: rephrase, maybe: the albedo feedback being absent in the simulation.

Fixed

l.43: maybe better: The first option is to use dedicated snow pack models coupled to RCMs...

Thanks for the suggestion. We have fixed it now in the revised version.

l.61: maybe: the model's performance => the model's behaviour

Fixed

10 l.77: please reformulate more carefully as LIG SMB is not known.

Done, the word "reproducing" is replaced with "simulating".

l.78: also here: I don't see how the advantage can be evaluated. Maybe "evaluate the effect..."

Corrected

sect. 2.1.1: Please highlight changes with respect to the earlier published model version.

In the revised version, we provide the differences compared to Zolles et al., 2021 in Section 2.1.1.

sect. 2.3: Maybe include a table of all experiments with some climte characteristics (mean JJA temperature and insolation range) and a figure of the topographies used (15km, 40km, T21).

For Section 2.3, a new table 1 with the characteristics of each experiment is included. Figures of topography for MAR, iLOVECLIM NH40, and SH40 as well as the native T21 grid, are also included (Fig. 2 and 3).

Fig 2: Maybe include sublimation,refeezing and melt in the pannels in the middle. Increase font size.

As we want to put more weight on the LIG part, the plots of differences in albedo, melt, refreezing, runoff, and sublimation between BESSI-MAR, ITM-MAR and MAR are in the Supplement.

l.290ff: a bit clumsy, please rephrase.

This sentence is removed as the comparison of MAR and iLOVECLIM in terms of climate forcing is put in the Supplement to support the results of Sect. 3.1 and 3.2.

table 2: Typo: "Greeland", maybe also include bias corrected BESSI.

The table is now replaced by Fig. 8.

The bias correction part is now included in the main text with the results of SMB of BESSI and ITM standalone with iLOVECLIM forcing before and after bias correction for both ice sheets (Sect 3.2 and 3.3.2).

l. 305: the "negtive SMB zone" is somewhat quite comparable to the more extensive ablation zone in MAR. Please discuss.

This is due to the bias correction impact on the crad included in ITM when running coupled to iLOVECLIM (Quiquet et al., 2024). In the new results of ITM stand-alone, we can see that the ablation zone in the southwest of Greenland is linked to temperature and short-wave radiation.

l. 324: "North of AIS" is no good orientation here, maybe Eastern Weddell Sea sector.

Fixed.

l. 355: avoid "overestimation" because this would imply that BESSI serves as the reference here- rather use something like "more sensitive than"

We have improved the word choices in the revised version.

Figure 9, upper panel: maybe include mean summer temperature for the two ice sheets.

Instead of the mean summer temperature for the two ice sheets, we now included a figure of temperature simulated on the ice sheet compared to the proxy (NGRIP and EPICA Dome).

l. 365: albedo feedback should be discussed in greater detail.

The albedo feedback is discussed in the Discussion as follows: "The albedo in ITM is fixed at 0.85, which is the value of ice grid points in iLOVECLIM, to give a clean comparison to BESSI. This can also be the reason behind the low runoff simulation in ITM-iLOVECLIM during the LIG. A lower albedo value, which means more solar radiation is considered, can increase the simulated runoff rate of ITM (Supplement Fig. S12b). However, using only one albedo value for the whole ice sheet is not realistic. ITM with a range of albedo for different altitudes and locations can provide satisfied results as in Quiquet et al. (2021)."

l. 374-380: a bit confusing, please rephrase.

The writing of this part is polished in the revised version.

l. 400: correct: south-western part of the GrIS

Done

l. 406: "cheaper cost" specify the computational cost for orientation (e.g. wall clock time/100 model years) of BESSI-iLOVECLIM and ITM-iLOVECLIM, maybe here, maybe somewhere else...

We are sorry for the missing information. The information is now mentioned in the Discussion.

l. 415: more processes do not always increase reliability- additional, poorly constrained feedbacks might actually increase uncertainty...

We agree with the reviewer. BESSI might not be reliable as its model structures are simplified. The simulation of various processes and feedback in BESSI does have some limitations. However, they are in the acceptable range as the model exhibits good results compared to MAR in the calibration/validation. Also, compared to ITM, which needs to be tuned for different climate conditions and tends to overestimate the runoff rate, BESSI is still more favorable to simulate SMB for transient paleo studies while maintaining a low computational cost.

30 l. 420: insolation is a common forcing for both BESSI and ITM.

Yes, sorry for the mistake.

l. 446-447: check grammar.

The sentence is rewritten as "The current SMB scheme of iLOVECLIM needs to be retuned for different climate forcings and study periods, which is not ideal for application in paleo studies."

Fig. B2: Caption does not seem to belong here.

We are sorry for the misplacing of the Appendix figures. It is fixed in the revised version.

Fig 1,2,4,5,6,7,8...: Increase font size.

All the figures are increased in font size in the revised version.

References

[revised manuscript text omitted]

---

## Referee Report (RR1)

**Second round of review of *'Using:a multi-layer snow model for transient paleo studies: surface mass balance evolution during the Last Interglacial'* by Hoang et al.**

I would like to thank the authors for the revised paper, which substantially improves on the original manuscript. I really appreciate the effort made in making it a more scientific paper with a widely extended analysis of the SMB evolution of the Greenland and Antarctic ice sheets over the last interglacial. Also, the methodology and the modelling framework and setup are much clearer now. I only have a few minor comments which should be addressed before the paper is acceptable for publication.

The text would still greatly benefit from a read by a native English speaker.

The line numbers in the comments below refer to the revised paper version with tracked changes.

**Minor comments**

L. 19-20: why would this improvement only be applicable to 'paleo periods'?

L. 108: 'in exponential relationship with' -> 'exponentially after'

L. 110: 'resulted' -> 'results'

Fig. 2: the caption should be extended to explain the different grids and what they represent

Equation 1: what are the units of m_runoff?

L. 241: 'during' -> 'for'

L. 242: Specify that sublimation is ignored. Maybe also consider writing out the SMB equation explicitly again.

Table 1: Specify which years are considered in the present-day case. Specify that Paleo study data are from iLOVECLIM. Why is the mean summer temperature over Antarctica so different in iLOVECLIM for present-day and PI?

L. 292: 'is' -> 'are'

L- 296-297: To quantify the impact of biases, you need to run simulations both with and without bias correction, no?

L. 524: in terms of timing yes, but can anything be said about the magnitude?

L. 699: 'satisfied' -> 'satisfying'

L. 729: check this sentence, sounds weird

L. 729: Possibly add some information on how much of the computation time would be consumed by BESSI in a fully coupled iLOVECLIM with GRISLI setup. Would it really be the bottleneck?

L. 752-753: Do you mean stronger sensitivity to model biases? It seems to me that ITM is responding less to LIG forcings than BESSI.

L. 875: why is the range different for RH? It seems to contradict the values larger than one in Fig. C2. Also, in the text it is mentioned that humidity is strongly underestimated in iLOVECLIM, but from Fig. C1 and C2 it seems to be overestimated? Is it because in one case you are referring to specific and in the other to relative humidity?

Generally, a comment on the large SW radiation biases in iLOVECLIM would help to interpret the results. How can SW radiation be almost a factor 10 too low over the ice sheets in the model? I presume this cannot be solely explained by differences in surface albedo, but is somehow related to clouds…? Is the bias larger than a factor 10 (which is the maximum used in the bias correction procedure) during the LIG? Could this (partly) explain the underestimation of melt in the ITM compared to BESSI?

---

## Author Response (AR2)

**RESPONSE TO REFEREE #1**

**In the following, the reviewer's comments are in blue, and our response is in black.**

**Second round of review of 'Using a multi-layer snow model for transient paleo studies: surface mass balance evolution during the Last Interglacial' by Hoang et al.**

I would like to thank the authors for the revised paper, which substantially improves on the original manuscript. I really appreciate the effort made in making it a more scientific paper with a widely extended analysis of the SMB evolution of the Greenland and Antarctic ice sheets over the last interglacial. Also, the methodology and the modelling framework and setup are much clearer now. I only have a few minor comments which should be addressed before the paper is acceptable for publication.

The text would still greatly benefit from a read by a native English speaker.

The authors thank referee #1 for taking the time and appreciating our efforts. Following your comments, we have corrected the grammar errors and improved the word use in the manuscript. In addition, we have also done our best to improve the quality of the manuscript in terms of English writing. We would like to benefit from the publisher's editing services once the manuscript is accepted.

The line numbers in the comments below refer to the revised paper version with tracked changes.

Minor comments

L. 19-20: why would this improvement only be applicable to 'paleo periods'?

We agree with the reviewer. It is not limited to paleo studies. We intended to emphasize the advantages of using iLOVECLIM, which is high computational efficiency, for multi-millennial simulations. We have modified the abstract with this in mind.

L. 108: 'in exponential relationship with' -> 'exponentially after'

Corrected.

L. 110: 'resulted' -> 'results'

Corrected.

Fig. 2: the caption should be extended to explain the different grids and what they represent

We apologize for the missing information. The caption is corrected with the appropriate information as: "Topography of iLOVECLIM for different resolutions: **(a)** NH40, **(b)** NH40 zoomed in Greenland with the red contour indicates present-day ice sheet extent, **(c)** SH40, **(d)** T21 with similar projection as NH40, **(e)** Similar to (d) but zoomed in Greenland and **(f)** T21 with similar projection as SH40."

Equation 1: what are the units of m_runoff?

Thank you for pointing this out. The unit of m_runoff is mWE. $d^{-1}$ and is added to the manuscript.

L. 241: 'during' -> 'for'

Fixed.

L. 242: Specify that sublimation is ignored. Maybe also consider writing out the SMB equation explicitly again.

Done.

Table 1: Specify which years are considered in the present-day case. Specify that Paleo study data are from iLOVECLIM. Why is the mean summer temperature over Antarctica so different in iLOVECLIM for present-day and PI?

We have added more detailed information about the forcings to the table caption and content.

About the mean summer temperature of iLOVECLIM over Antarctica, the presented data is boreal summer (June-July-August) for PI and LIG experiments, while the data for the present-day experiment is from December-January-February. We apologize for the inconsistency. The data has been updated with the correct data (DJF). In addition, the captions of Table 1 and figures with summer temperature (Fig. S4 and S7) have been updated with details on the summer months.

L. 292: 'is' -> 'are'

Fixed.

L- 296-297: To quantify the impact of biases, you need to run simulations both with and without bias correction, no?

This is what we have done, but maybe the sentence is not clear. We have adjusted it as follows: "To quantify the impact of these biases on the SMB simulation, in addition to original climate forcings, we also run BESSI and ITM with the bias-corrected version of iLOVECLIM."

L. 524: in terms of timing yes, but can anything be said about the magnitude?

We have modified the text of Fig.10d as follows: "For Antarctica, the comparison of the simulated local surface temperature of iLOVECLIM and temperature change proxy at EPICA Dome C (EDC) is presented in Fig. 10d. The change in the simulated temperature shows a good agreement with the proxy-based data regarding timing. However, the warming at EDC during the LIG compared to PI in our work is only 0.59ºC, while the value suggested by Jouzel et al. (2007) is about 4.5ºC. This difference might result from the fixed ice sheet mask and topography in our simulations. It is possible that the West Antarctic Ice Sheet was smaller during the LIG, leading to a change in surface elevation and ice extent. This can, in turn, increase the temperature at EDC. This part of warming is not taken into account in our simulations."

L. 699: 'satisfied' -> 'satisfying'

Done

L. 729: check this sentence, sounds weird

L. 729: Possibly add some information on how much of the computation time would be consumed by BESSI in a fully coupled iLOVECLIM with GRISLI setup. Would it really be the bottleneck?

BESSI is not yet coupled to iLOVECLIM-GRISLI so it is not certain to give the prediction of the computation cost once it is done.

This part has been adjusted as follows: "Replacing ITM with BESSI to provide SMB to the ice sheet model GRISLI in iLOVECLIM framework can produce more physical results. Nonetheless, BESSI requires more input variables than ITM, making it more sensitive to certain biases in iLOVECLIM, such as humidity. BESSI is also more computationally expensive (30 years per minute for the T21 grid) than a parameterization like ITM. However, considering the computational cost of iLOVECLIM (500 years per day), the extra cost of having BESSI instead of ITM in the framework is relatively small. In addition, we can be more confident in its response to a change in climate since it explicitly simulates many processes, unlike ITM."

L. 752-753: Do you mean stronger sensitivity to model biases? It seems to me that ITM is responding less to LIG forcings than BESSI.

We have corrected the sentences as: "Notably, the comparison between BESSI and ITM during the Last Interglacial suggests a stronger sensitivity of ITM to the biases in the climate forcings".

L. 875: why is the range different for RH? It seems to contradict the values larger than one in Fig. C2. Also, in the text it is mentioned that humidity is strongly underestimated in iLOVECLIM, but from Fig. C1 and C2 it seems to be overestimated? Is it because in one case you are referring to specific and in the other to relative humidity?

We apologize for the confusion. The text has been adjusted as follows: "In order to avoid extreme value, the ratio $\frac{\overline{X_{ERA5}}}{\overline{X_{\iota LC}}}$ is limited to be in the range of 0.1 to 10.0. In addition, for relative humidity only, once the bias is corrected, the value is restricted between 0.15 and 1.0 (15-100\%) to avoid unrealistic values."

The color bars of Fig. C1 and C2 have also been adjusted to illustrate better the range of the bias correction factors. These figures indicate an underestimation of relative humidity in the center North of Greenland and the interior of Antarctica. Particularly, the ranges of the ratio of relative humidity between ERA5 and iLOVECLIM are about 1.1-1.5 for Greenland and up to 5.0 for Antarctica. These numbers indicate a substantial underestimation of relative humidity in Antarctica, considering that the normal range is mostly around 15 to 100 %. Similarly, when compared to MAR, iLOVECLIM also underestimates relative humidity, as shown in Fig. S4 and S7.

Generally, a comment on the large SW radiation biases in iLOVECLIM would help to interpret the results. How can SW radiation be almost a factor 10 too low over the ice sheets in the model? I presume this cannot be solely explained by differences in surface albedo, but is somehow related to clouds…? Is the bias larger than a factor 10 (which is the maximum used in the bias correction procedure) during the LIG? Could this (partly) explain the underestimation of melt in the ITM compared to BESSI?

We also noticed the unexpectedly low range of shortwave radiation. The reasons behind this can be the prescribed cloud and fixed vertical radiation scheme inside of iLOVECLIM. Indeed, this problem should be further investigated in future work. A comment about the biases of iLOVECLIM is added to the main text to provide more information.

It is uncertain whether the bias correction factors exceed 10 during the LIG as there are no data for this period. In this work, the bias correction factors are obtained from the present-day climate (1979-2021) and applied to the LIG, assuming the biases are constant with time. However, for the present day, the bias correction factors can be larger than 10, particularly for total precipitation and shortwave radiation. To test the impact of the threshold value, we have run BESSI and ITM with bias-corrected iLOVECLIM during the period of 1979-2021 with no maximum threshold on the bias correction factor of shortwave radiation. The changes in annual mean SMB in BESSI-iLOVECLIM and ITM-iLOVECLIM between with and without threshold for the bias correction factor of shortwave radiation are relatively small, and the SMB patterns remain unchanged, as shown in Fig. R1 and R2. We also do the same test for the time slice 128.5 kaBP when the SMB reaches its minimum value during the LIG. The results of this time slice are similar to the present-day run, as shown in Fig. R3 and R4. For Antarctica, we can see the grid cells with high differences, but the magnitude of SMB change is still relatively small (0.5% for BESSI-iLOVECLIM and 1% for ITM-iLOVECLIM).

Hence, the threshold of the bias correction factors is unlikely to impact the melt simulation in ITM.

[Figure]

**Figure RA1**. Comparison of present-day annual mean SMB from bias-corrected BESSI-iLOVECLIM and ITM-iLOVECLIM for experiments with a threshold (10 maximum) **(a)** not applied and **(b)** applied to the bias correction factors of the shortwave radiation.

[Figure]

**Figure RA2**. Similar to Figure RA1 but for Antarctica.

[Figure]

**Figure RA3**. Similar to Figure RA1 but for 128.5 kaBP.

[Figure]

**Figure RA4**. Similar to Figure RA2 but for 128.5 kaBP.

*References*

Jouzel, J., Masson-Delmotte, V., Cattani, O., Dreyfus, G., Falourd, S., Hoffmann, G., Minster, B., Nouet, J., Barnola, J.M., Chappellaz, J., Fischer, H., Gallet, J.C., Johnsen, S., Leuenberger, M., Loulergue, L., Luethi, D., Oerter, H., Parrenin, F., Raisbeck, G., Raynaud, D., Schilt, A., Schwander, J., Selmo, E., Souchez, R., Spahni, R., Stauffer, B., Steffensen, J.P., Stenni, B., Stocker, T.F., Tison, J.L., Werner, M., Wolff, E.W., 2007. Orbital and Millennial Antarctic Climate Variability over the Past 800,000 Years. Science 317, 793–796. https://doi.org/10.1126/science.1141038

**RESPONSE TO REFEREE #2**

**In the following, the reviewer's comments are in blue, and our response is in black**

Thank you to the author, they have made quite an effort to implement the recommendations by both reviewers. I am mostly satisfied with the revised version, and have only three small concerns:

The authors thank referee #2 for taking the time and appreciating our efforts. We have corrected the manuscript following your comments. The detailed answers are in the following.

Table 1: Please check mean summer shortwave radiation, the iLoveclim values are completely off and I suspect this is not the same variable than for MAR, also is summer insolation in the lower part of the table the same as mean summer shortwave radiation? Maybe use the same variable/name in both cases

We apologize for the confusion.

For the mean summer shortwave radiation, we computed the mean value of the summer months (June-July-August for Greenland and December-January-February for Antarctica) on the corresponding present-day ice sheet extent. The shortwave radiation in iLOVECLIM is lower than in MAR due to the biases of this variable (as shown in Fig. C1 and C2). This issue, together with the biases of relative humidity in Antarctica, will be further investigated in future works.

For the Last Interglacial simulation, the summer insolation refers to the summer insolation at 65$^{\circ}$N for Greenland and 65$^{\circ}$S for Antarctica.

The table's caption is adjusted to provide more information for the readers as: "Climate characteristics of two different climate forcings: MAR and iLOVECLIM for different experiments. The calibration/validation is carried out from 1979 to 2021 with forcings from MAR and iLOVECLIM. Mean summer shortwave radiation and mean summer temperature are calculated based on the present-day ice sheet extent. The climate forcings for the Last Interglacial (LIG) comes from iLOVECLIM only. The summer insolation of the paleo study corresponds to the summer insolation of 65$^{\circ}$N for the Greenland Ice Sheet (GrIS) and 65$^{\circ}$S for the Antarctic Ice Sheet (AIS). The summer months are June-July-August for GrIS and December-January-February for AIS."

Fig.8: the bold color is used for the bias corrected forcing of BESSI and for the non-bias corrected of ITM, using bold colors for both bias corrected forcings would be more intuitive.

We have adjusted the color choices as recommended.

Fig. 10 caption should state that isotope values are from ice cores and distinguish whether time series reflect global or local temperatures. Also the text describing Fig. 10 could be more clear in this, maybe also mention effect of height changes for local temperatures.

Thank you for your useful comments.

We have adjusted the caption of Fig. 10 with more description. We now indicate that Fig. 10c-d are local surface temperatures and mention that the proxy data includes the impact of elevation changes while we do not in our simulations. The corresponding text has also been adjusted.

---

## Author Response (AR3)

**RESPONSE TO EDITOR**

In the following, the Editor's comments are in blue, and our response is in black.

Thank you for submitting your revised manuscript. You have appropriately answered the Reviewers' comments, however I wonder if there is still an outstanding issue with your shortwave radiation plots on figures S4 and S7. Maybe the issue is that you are comparing net shortwave radiation to downward shortwave radiation? Can you please double check this and amend as needed?

We sincerely would like to thank two referees for noticing the problem relating to shortwave radiation and the Editor for insisting on a more detailed check.

We have conducted a thorough review of the ECBilt code and found that indeed there was a mislabeled text field regarding shortwave radiation in the outputs. We sincerely apologize for not noticing this earlier.

We have now corrected the iLOVECLIM forcings accordingly and re-run all the simulations of BESSI and ITM. As expected, the simulated SMB by both models is now much lower for both ice sheets with the correct shortwave radiation for the present-day condition and pre-industrial period. These results are consistent with the warm temperature bias (+10°C) of iLOVECLIM. After correcting the biases, both models' results are in a similar range as MAR.

We have updated the text and figures of the manuscript with new results.

Although this is a major correction, fortunately, it does not change the main outcomes of the paper:

1. BESSI-MAR provides decent results when compared to MAR and contrary to the ITM-MAR.
2. BESSI is physics-based and does not need retuning.
3. BESSI strongly depends on the quality of the forcing, and the biases of iLOVECLIM strongly affect the model's results.
4. The sensitivity of BESSI to the climate forcings of the Last Interglacial (LIG) is higher than ITM.

Once again, we apologize for this mistake.